# Landscape mapping of shared antigenic epitopes and their cognate TCRs of tumor-infiltrating T lymphocytes in melanoma

Kenji Murata[1†], Munehide Nakatsugawa[1†], Muhammed A Rahman[1], Linh T Nguyen[1], Douglas G Millar[1], David T Mulder[1,2], Kenji Sugata[1], Hiroshi Saijo[1], Yukiko Matsunaga[1], Yuki Kagoya[1], Tingxi Guo[1], Mark Anczurowski[1], Chung-Hsi Wang[1,3], Brian D Burt[1], Dalam Ly[1], Kayoko Saso[1], Alexandra Easson[4], David P Goldstein[4], Michael Reedijk[4,5,6], Danny Ghazarian[7], Trevor J Pugh[2,5,8], Marcus O Butler[1,3,6,9], Tak W Mak[1,3,6], Pamela S Ohashi[1,3,6], Naoto Hirano[1,3,6,8]*

[1]Tumor Immunotherapy Program, Campbell Family Institute for Breast Cancer Research, Princess Margaret Cancer Centre, University Health Network, Toronto, Canada; [2]Princess Margaret Cancer Centre, University Health Network, Toronto, Canada; [3]Department of Immunology, University of Toronto, Toronto, Canada; [4]Department of Surgical Oncology, University Health Network, Toronto, Canada; [5]Department of Medical Biophysics, University of Toronto, Toronto, Canada; [6]Campbell Family Institute for Breast Cancer Research, Princess Margaret Cancer Centre, University Health Network, Toronto, Canada; [7]Department of Laboratory Medicine and Pathobiology, University Health Network, Toronto, Canada; [8]Ontario Institute for Cancer Research, Toronto, Canada; [9]Department of Medicine, University of Toronto, Toronto, Canada

*For correspondence:
naoto.hirano@uhnresearch.ca

[†]These authors contributed equally to this work

**Abstract** HLA-restricted T cell responses can induce antitumor effects in cancer patients. Previous human T cell research has largely focused on the few HLA alleles prevalent in a subset of ethnic groups. Here, using a panel of newly developed peptide-exchangeable peptide/HLA multimers and artificial antigen-presenting cells for 25 different class I alleles and greater than 800 peptides, we systematically and comprehensively mapped shared antigenic epitopes recognized by tumor-infiltrating T lymphocytes (TILs) from eight melanoma patients for all their class I alleles. We were able to determine the specificity, on average, of 12.2% of the TILs recognizing a mean of 3.1 shared antigen-derived epitopes across HLA-A, B, and C. Furthermore, we isolated a number of cognate T cell receptor genes with tumor reactivity. Our novel strategy allows for a more complete examination of the immune response and development of novel cancer immunotherapy not limited by HLA allele prevalence or tumor mutation burden.

## Introduction

Cancer immunotherapy is a cancer treatment that is designed to harness the power of the body's immune system to fight cancer (**Finn, 2018a**). The magnitude and breadth of T cell responses can serve as a critical biomarker in cancer immunotherapy and conventional cancer treatments, such as chemotherapy and radiation (**Schumacher et al., 2015**). Immunotherapy is now widely considered a

**eLife digest** The immune system is the body's way of defending itself, offering protection against diseases such as cancer. But to remove the cancer cells, the immune system must be able to identify them as different from the rest of the body. All cells break down proteins into shorter fragments, known as peptides, that are displayed on the cell surface by a protein called human leukocyte antigen, HLA for short. Cancer cells display distinctive peptides on their surface as they generate different proteins to those of healthy cells. Immune cells called T cells use these abnormal peptides to identify the cancer so that it can be destroyed.

Sometimes T cells can lack the right equipment to detect abnormal peptides, allowing cancer cells to hide from the immune system. However, T cells can be trained through a treatment called immunotherapy, which provides T cells with new tools so that they can spot the peptides displayed by HLA on the previously 'hidden' cancer cells.

There are many different forms of HLA, each of which can display different peptides. Current research in immunotherapy commonly targets only a subset of HLA forms, and not all cancer patients have these types. This means that immunotherapy research is only likely to be of most benefit to a limited number of patients. Immunotherapy could be made effective for more people if new cancer peptides that are displayed by the other 'under-represented' forms of HLA were identified.

Murata, Nakatsugawa et al. have now used T cells that were taken from tumors in eight patients with melanoma, which is a type of skin cancer. A library of fluorescent HLA-peptides was generated – using a new, simplified methodology – with 25 forms of HLA that displayed over 800 peptides. T cells were then mixed with the library to identify which HLA-peptides they can target. As a result, Murata, Nakatsugawa et al. found the cancer targets of around 12% of the tumor-infiltrating T cells tested, including those from under-represented forms of HLA. Consequently, these findings could be used to develop new immunotherapies that can treat more patients.

game-changer, as it is rapidly becoming the 4th pillar of cancer treatment, following surgery, radiotherapy, and chemotherapy.

The adoptive transfer of antitumor T cells can induce clinical responses in cancer patients (*Feldman et al., 2015*; *Hu et al., 2018*; *Karpanen and Olweus, 2015*; *Rapoport et al., 2015*; *Robbins et al., 2015*; *Sahin and Türeci, 2018*; *Tran et al., 2016*; *Tran et al., 2017*). However, comprehensive analysis of the specificity of antitumor T cell responses against mutated or non-mutated tumor antigens is lacking. This deficit is exacerbated by the fact that non-mutated antigens are greater in number than mutated antigens by multiple orders of magnitude (*Finn and Rammensee, 2018*) and the high polymorphism of HLA genes (*Shao et al., 2018*). The clinical importance of identifying antitumor T cell responses is highlighted by the results of recent clinical trials which attest that a shared antigen, such as NY-ESO-1, can be targeted with potent on-target efficacy and minimal off-target toxicity in T cell receptor (TCR) gene therapy (*Rapoport et al., 2015*; *Robbins et al., 2015*; *Thomas et al., 2018*). Unlike shared antigens, the vast majority of neoantigens are unique to each patient (*Coulie et al., 2014*; *Finn, 2018b*). The elucidation of T cell epitopes derived from shared antigens may facilitate the robust development of an efficacious and safe adoptive T cell therapy that is readily available to a larger cohort of cancer patients.

Analysis of antigen-specific T cells using peptide/HLA (pHLA) multimers has been utilized as a standard technique in immunology over the past two decades (*Bentzen and Hadrup, 2017*; *Dey et al., 2019*; *Dolton et al., 2015*). Recent progress in multimer technology has enabled the high-throughput analysis of antigen-specific T cells during immune responses (*Bentzen and Hadrup, 2017*). However, the production of each multimer is still cumbersome and expensive because in vitro peptide exchange of generated complete pHLA proteins requires multiple complicated steps (*Toebes et al., 2006*) and some class I alleles are not easily produced in bacteria (*Migueles et al., 2000*). Therefore, further advancements in multimer technology are needed to investigate the multitude of potential tumor-associated antigen (TAA) peptides presented by any given class I allele.

The combination of pHLA multimer-based analysis and functional assays are utilized to measure antigen-specific T cell responses. We have developed a novel peptide-exchangeable pHLA class I

multimer technology that can be applied for any given class I allele and bound peptide. Unlike comparable methods (*Andersen et al., 2012*; *Migueles et al., 2000*), our novel pHLA multimer technology enables a one-step peptide exchange in one tube and is more effective in class I alleles that are difficult to produce in bacteria. These technical advantages allow for a rapid, versatile, and less labor-intensive examination of the immune response. In addition, to functionally detect HLA-restricted antigen-specific T cell responses, we previously demonstrated the versatility of gene-engineered artificial antigen-presenting cells (APCs) individually expressing various class I alleles as a single HLA allele (*Butler and Hirano, 2014*). Our strategy using a paired library of the novel pHLA multimer and human cell-based artificial APC system enables identification of novel class I epitopes and detection of cognate T cells in a comprehensive and systematic way.

## Results

### High-throughput staining of polyclonally expanded TILs with a panel of pHLA multimers

The adoptive transfer of TILs can induce sustained antitumor responses in patients with metastatic melanoma in combination with lymphodepletion and IL-2 administration (*Yang and Rosenberg, 2016*). It is well established that melanoma TILs contain antitumor T cells that are specific for both non-mutated and mutated antigens (*Andersen et al., 2012*; *Bentzen et al., 2016*; *Duhen et al., 2018*; *Ye et al., 2014*). Here, TILs were isolated from eight metastatic melanoma patients, polyclonally expanded in vitro (*Nguyen et al., 2019*), and then examined for their shared antigen specificities recognizing epitopes in the context of all 25 different class I alleles present. The combination of structure-based analysis using pHLA multimers and functional analysis has been widely used to measure antigen-specific T cell responses (*Bentzen and Hadrup, 2017*; *Bentzen et al., 2016*; *Newell et al., 2013*). We initially stained the T cells using our novel peptide-exchangeable pHLA multimers with bound peptides that were previously known or predicted by publicly available algorithms (*Andreatta and Nielsen, 2016*). The previously known peptides were chosen from the Cancer Antigenic Peptide Database (https://caped.icp.ucl.ac.be) for all the class I alleles of eight melanoma patients, and we also synthesized several high-affinity binding peptides derived from TAAs based on prediction algorithms for the class I alleles that were positive in multiple patients (*Figure 1*, *Figure 1—figure supplement 1*, *Supplementary file 1*). As shown in *Table 1*, all eight samples showed positivity for at least one of the chosen multimers. In addition to the previously known epitopes, we found that the newly predicted epitopes $A*02:01/ABCB5_{700-708}$ and $C*05:01/tyrosinase_{460-468}$ were immunogenic (*Figure 1A,B*, *Figure 2A–C*).

### All the multimer-positive melanoma TILs are functional

All the multimer-positive T cells secreted detectable IFN-γ in an HLA-restricted peptide-specific manner according to ELISPOT analysis, with the exception of $A*02:01/gp100_{154-162}$ and $A*02:01/tyrosinase_{369-377}$ T cells among the M87 TILs, only 0.14% and 0.23% of which were stained with multimer, respectively (*Figure 2A,B*, *Figure 2—figure supplement 1A,B,D*). However, stimulation of M87 TILs with A*02:01-artificial APCs engineered to express the A*02:01 gene as a single HLA allele and pulsed with the $gp100_{154-162}$ or $tyrosinase_{369-377}$ peptide resulted in expansion and IFN-γ production in an $A*02:01/gp100_{154-162}$- or $A*02:01/tyrosinase_{369-377}$-specific manner, respectively, validating the low percentage of staining (*Figure 2D,E*). Notably, only a single nominal stimulation was performed which was sufficiently weak to induce the expansion of in vivo-primed peptide-specific T cells, but avoid the in vitro priming and deletion of high-avidity T cells (*Hirano et al., 2006*). The low percentage of staining of M37 and M87 TILs with A*24:02/gp100-intron4 (0.19%) and $C*05:01/tyrosinase_{460-468}$ (0.54%) multimers, respectively, was similarly validated; one nominal peptide-specific stimulation of M37 and M87 TILs with the HLA class I-matched artificial APCs led to their expansions in an A*24:02/gp100-intron4- and $C*05:01/tyrosinese_{460-468}$-specific manner, respectively (*Figure 2C*, *Figure 2—figure supplement 1C*).

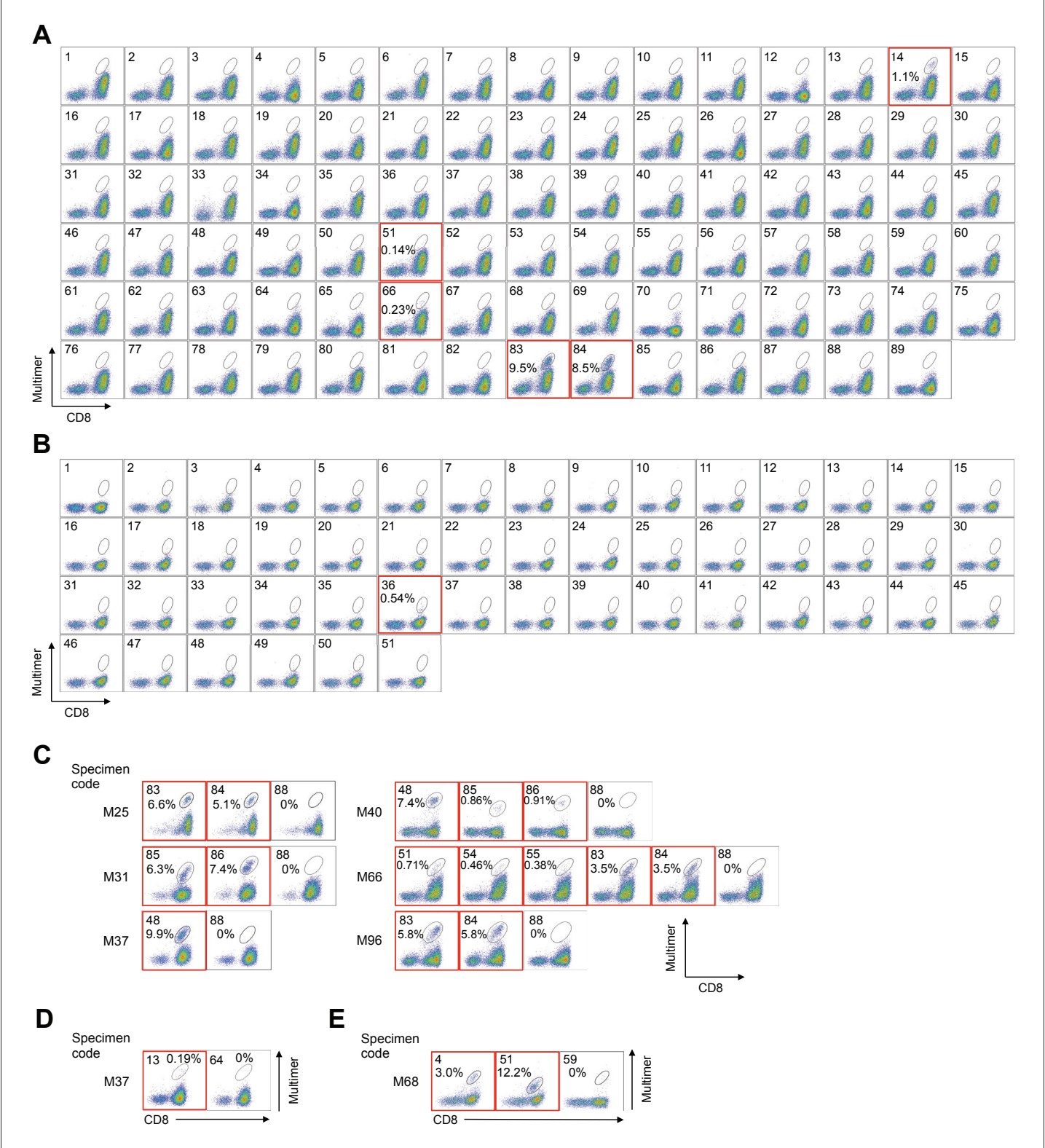

**Figure 1.** High−throughput staining of polyclonally expanded TILs with a panel of pHLA multimers. (**A, B**) Staining of HLA-A*02:01[+] (**A**) and C*05:01[+] (**B**) M87 TILs with pHLA multimers whose peptides were previously known or predicted by publicly available algorithms. The peptides employed for the A*02:01 and C*05:01 multimers are shown in ***Supplementary files 1b and f***, respectively. (**A**) The TILs showed positivity to the A*02:01/ABCB5$_{700-708}$ (No. 14), A*02:01/gp100$_{154-162}$ (No. 51), A*02:01/tyrosinase$_{369-377}$ (No. 66), A*02:01/wild-type MART1$_{27-35}$ (No. 83), and A*02:01/heteroclitic MART1$_{26-35}$ (No. 84) multimers. (**B**) The TILs showed positivity to the C*05:01/tyrosinase$_{460-468}$ (No. 36) multimer. The A*02:01/HIV pol$_{476-484}$ (No. 87 in A), A*02:01/

*Figure 1 continued on next page*

Figure 1 continued

HTLV-1 tax$_{11-19}$ (No. 88 in A), A*02:01/unexchanged (No. 89 in A), C*05:01/HIV rev$_{67-75}$ (No. 50 in B), and C*05:01/unexchanged (No. 51 in B) multimers were used as negative controls. (C–E) Positive staining of TILs with a panel of pHLA multimers for HLA-A*02:01$^+$ M25, M31, M37, M40, M66, and M96 TILs (C), HLA-A*24:02$^+$ M37 TILs (D), and HLA-B*07:02$^+$ M68 TILs (E). All the high-throughput staining data are shown in *Figure 1—figure supplement 1*. (C) M25 and M96 TILs showed positivity to the A*02:01/wild-type MART1$_{27-35}$ (No. 83) and A*02:01/heteroclitic MART1$_{26-35}$ (No. 84) multimers. M31 TILs showed positivity to the A*02:01/wild-type NY-ESO-1$_{157-165}$ (No. 85) and A*02:01/heteroclitic NY-ESO-1$_{157-165}$ (No. 86) multimers. M37 TILs showed positivity to the A*02:01/SSX2$_{41-49}$ (No. 48) multimer. M40 TILs showed positivity to the A*02:01/SSX2$_{41-49}$ (No. 48), A*02:01/wild-type NY-ESO-1$_{157-165}$ (No. 85), and A*02:01/heteroclitic NY-ESO-1$_{157-165}$ (No. 86) multimers. M66 TILs showed positivity to the A*02:01/gp100$_{154-162}$ (No. 51), A*02:01/gp100$_{209-217}$ (No. 54), A*02:01/gp100$_{280-288}$ (No. 55), A*02:01/wild-type MART1$_{27-35}$ (No. 83), and A*02:01/heteroclitic MART1$_{26-35}$ (No. 84) multimers. (D) M37 TILs showed positivity to the A*24:02/gp100-intron4 (No. 13) multimer. (E) M68 TILs showed positivity to the B*07:02/NY-ESO-1$_{60-72}$ (No. 4) and B*07:02/MAGE-A1$_{289-297}$ (No. 51) multimers. The A*02:01/HTLV-1 tax$_{11-19}$ (No. 88 in C), A*24:02/HTLV-1 tax$_{301-309}$ (No. 64 in D), and B*07:02/HIV nef$_{128-137}$ (No. 59 in E) multimers were used as negative controls. The red squares highlight positive TIL staining with pHLA multimers. The percentage of multimer$^+$ cells in CD8$^+$ T cells is shown. The data shown are representative of two independent experiments.

The online version of this article includes the following figure supplement(s) for figure 1:

**Figure supplement 1.** High-throughput staining of polyclonally expanded TILs with a panel of HLA-A*02:01, A*24:02, and B*07:02 multimers.

## Identification of low-frequency antigen-specific T cells following peptide-specific expansion

The melanoma TILs that were studied had been polyclonally expanded in the absence of peptide-specific stimulation using an established protocol (*Nguyen et al., 2019*). Conceivably, the expanded TILs contained a low (below the detection limit) frequency of T cells targeting shared antigens (*Bentzen et al., 2016*). To explore this possibility, TILs were weakly stimulated once with class I-matched artificial APCs pulsed with the peptides listed in *Supplementary file 1* (*Hirano et al., 2006*). The B*18:01/MAGE-A3$_{167-176}$ multimer positivity of polyclonally expanded M40 TILs was only 0.04% prior to peptide-specific stimulation. However, following one stimulation with B*18:01-artificial APCs pulsed with the MAGE-A3$_{167-176}$ peptide, 5.5% of the TILs were stained with the cognate multimer and secreted IFN-γ in a B*18:01/MAGE-A3$_{167-176}$-specific manner (*Figure 3A,B*).

## T cell epitope determination of in vitro-expanded TILs using overlapping peptides

Since pHLA multimer production requires the use of a peptide with a known exact sequence, it is not straightforward or practical to conduct high-throughput screening for new epitope peptides using a pHLA multimer-based strategy. To identify new epitope peptides, we conducted functional assays using artificial APCs, which can take up and process longer peptides and present epitope peptides via class I molecules, as stimulator cells (*Butler and Hirano, 2014*). The class I-matched artificial APCs were pulsed with overlapping peptides (20-mers with an overlap of 15 amino acids) to cover the whole proteins of six shared antigens (MART1, NY-ESO-1, SSX2, gp100, MAGE-A1, and survivin) that are frequently expressed by melanomas (*Finn, 2018b*; *Supplementary file 2*) and used as stimulators in cytokine ELISPOT assays.

When stimulated with B*18:01-artificial APCs pulsed with MART1-derived overlapping peptides, B*18:01$^+$ M87 TILs showed positive responses to two adjacent peptides with the shared sequence $_{21}$YTTAEEAAGIGILTV$_{35}$ (*Figure 4A*, *Supplementary file 2a*). Using a series of deletion mutant peptides, we determined the minimally required epitope peptide, $_{25}$EEAAGIGIL$_{33}$ presented by B*18:01 molecules. Notably, this epitope partially overlaps with but is distinct from one of the most immunogenic epitopes, A*02:01/MART1$_{27-35}$, suggesting that this region of the MART1 protein is an immunological hotspot (*Cole et al., 2010*; *Kawakami et al., 1994*). Importantly, the B*18:01/MART1$_{25-33}$ multimer successfully stained up to 9.2% of the polyclonally expanded M87 TILs, suggesting that the B*18:01/MART1$_{25-33}$ T cells were a dominant population of TILs (*Figure 4B,C*). Similarly, we detected C*03:04/NY-ESO-1$_{92-100}$ T cells, the frequency of which was 18.2% of polyclonally expanded M31 TILs, and they were also a dominant population of TILs (*Figure 4—figure supplement 1D–F*, *Supplementary file 2b*). Additionally, following peptide-specific stimulation with B*40:01-artificial APCs pulsed with NY-ESO-1-derived overlapping peptides, a novel B*40:01-restricted NY-ESO-1 epitope, $_{125}$EFTVSGNIL$_{133}$, was determined (*Figure 4D–F*, *Supplementary file 2b*). The B*40:01/NY-ESO-1$_{125-133}$ multimer positivity was only 0.16% in polyclonally expanded M31 TILs. However, following one peptide-specific stimulation of the TILs with B*40:01-artificial APCs

**Table 1.** Summary of shared antigenic epitopes recognized by melanoma TILs and their cloned TCRs.

| Specimen code | HLA-A | HLA-B | HLA-C | Shared antigenic epitopes | % of multimer-positive CD8+ TILs | IFN-$\gamma$ ELISPOT analysis | Number of cloned TCRs | Tumor reactivity |
|---|---|---|---|---|---|---|---|---|
| M25 | 01:01/ 02:01 | 07:02/ 08:01 | 07:01/ 07:02 | A*02:01/MART1$_{27-35/26-35}$ (wild-type/ heteroclitic) | 6.6/5.1 | +/+ | ND | ND |
| M31 | 02:01/ 24:02 | 40:01/ 44:02 | 03:04/ 05:01 | A*02:01/NY-ESO-1$_{157-165}$ (wild-type/ heteroclitic) | 6.3/7.4 | +/+ | ND | ND |
| | | | | B*40:01/NY-ESO-1$_{125-133}$ | 0.16 | + | 1 | + |
| | | | | B*40:01/gp100$_{448-458}$ | 0.11 | + | 1 | + |
| | | | | C*03:04/NY-ESO-1$_{92-100}$ | 18.2 | + | 1 | + |
| M37 | 02:01/ 24:02 | 14:02/ 48:01 | 08:02/ 08:03 | A*02:01/SSX2$_{41-49}$ | 9.9 | + | 2 | + |
| | | | | A*24:02/gp100-intron4 | 0.19 | + | 1 | + |
| M40 | 02:01/ 30:02 | 18:01/ 45:01 | 05:01/ 06:02 | A*02:01/SSX2$_{41-49}$ | 7.4 | + | 2 | ND |
| | | | | A*02:01/NY-ESO-1$_{157-165}$ (wild-type/ heteroclitic) | 0.86/0.91 | +/+ | ND | ND |
| | | | | B*18:01/MAGE-A3$_{167-176}$ | 0.04 | + | 1 | + |
| | | | | C*06:02/gp100$_{190-198}$ | 1.2 | + | 1 | + |
| M66 | 02:01/ 32:01 | 07:02/ 27:05 | 02:02/ 07:02 | A*02:01/gp100$_{154-162}$ | 0.71 | + | ND | ND |
| | | | | A*02:01/gp100$_{209-217}$ | 0.46 | + | ND | ND |
| | | | | A*02:01/gp100$_{280-288}$ | 0.38 | + | ND | ND |
| | | | | A*02:01/MART1$_{27-35/26-35}$ (wild-type/ heteroclitic) | 3.5/3.5 | +/+ | ND | ND |
| M68 | 03:01/ 24:02 | 07:02/- | 07:02/- | B*07:02/NY-ESO-1$_{60-72}$ | 3.0 | + | 1 | + |
| | | | | B*07:02/MAGE-A1$_{289-297}$ | 12.2 | + | 1 | + |
| M87 | 01:01/ 02:01 | 18:01/- | 05:01/ 07:01 | A*02:01/ABCB5$_{700-708}$ | 1.1 | + | 1 | ND |
| | | | | A*02:01/gp100$_{154-162}$ | 0.14 | + | ND | ND |
| | | | | A*02:01/tyosinase$_{369-377}$ | 0.23 | + | ND | ND |
| | | | | A*02:01/MART1$_{27-35/26-35}$ (wild-type/ heteroclitic) | 9.5/8.5 | +/+ | ND | ND |
| | | | | B*18:01/MART1$_{25-33}$ | 9.2 | + | 1 | + |
| | | | | C*05:01/tyrosinase$_{460-468}$ | 0.54 | + | 1 | + |
| | | | | C*07:01/gp100$_{479-487}$ | 0.14 | + | 1 | + |
| M96 | 01:01/ 02:01 | 08:01/ 51:01 | 07:01/ 15:02 | A*02:01/MART1$_{27-35/26-35}$ (wild-type/ heteroclitic) | 5.8/5.8 | +/+ | ND | ND |

ND; Not determined.

pulsed with the NY-ESO-1$_{125-133}$ peptide, the frequency of B*40:01/NY-ESO-1$_{125-133}$ T cells increased to 6.1% (*Figure 4E*). Using a similar strategy, we identified HLA-B*40:01/gp100$_{448-458}$, C*06:02/gp100$_{190-198}$, and C*07:01/gp100$_{479-487}$ T cells, which accounted for 0.11%, 1.2%, and 0.14% of CD8+ T cells among the polyclonally expanded TILs, respectively (*Figure 4—figure supplement 1A–C,G–L*, *Supplementary file 2d*). Following peptide-specific stimulation with class I-matched artificial APCs, the frequency of B*40:01/gp100$_{448-458}$ and C*07:01/gp100$_{479-487}$ T cells increased to 1.8% and 1.2%, respectively, excluding the possibility that the low percentages of staining represented false positives (*Figure 4—figure supplement 1B,K*). Although we performed similar experiments using overlapping peptides for SSX2, MAGE-A1, and survivin, no T cell response was observed. It should be noted that in this study, we did not investigate the expression of shared

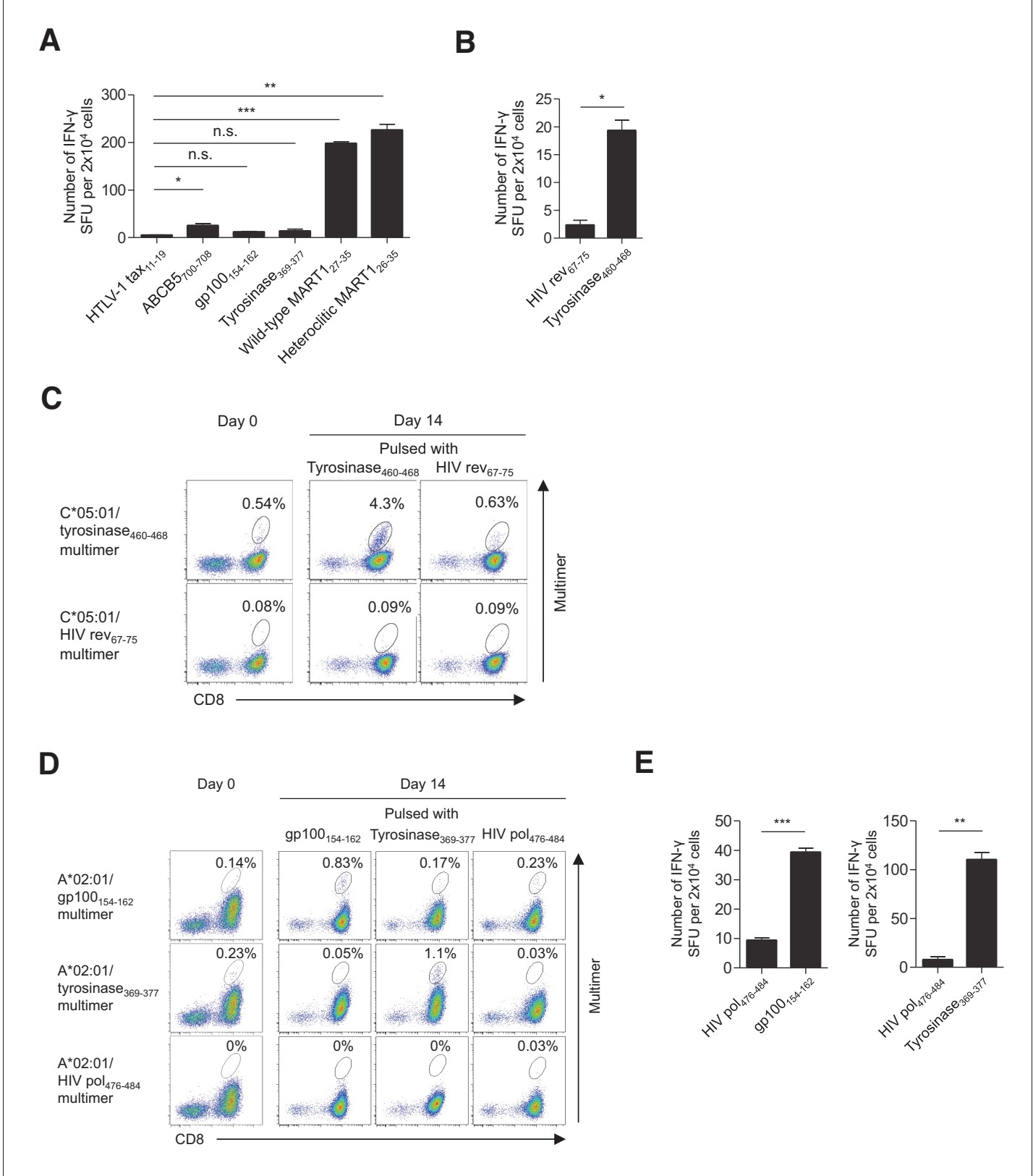

**Figure 2.** Functional assessment of multimer-positive melanoma TILs. (**A**) IFN-γ production by M87 TILs in an HLA-A*02:01-restricted peptide-specific manner. T2 cells pulsed with the indicated peptide were used as stimulator cells. The HTLV-1 tax$_{11-19}$ peptide was employed as a control. (**B**) IFN-γ production by C*05:01-positive M87 TILs in a C*05:01/tyrosinase$_{460-468}$-specific manner. C*05:01-artificial APCs pulsed with tyrosinase$_{460-468}$ or the HIV

*Figure 2 continued on next page*

*Figure 2 continued*

rev$_{67-75}$ (control) peptide were employed as stimulator cells. (C) Expansion of C*05:01/tyrosinase$_{460-468}$ T cells in M87 TILs in an HLA-restricted peptide-specific manner. The M87 TILs were stimulated with C*05:01-artificial APCs pulsed with the indicated peptide. Data for the C*05:01/tyrosinase$_{460-468}$ multimer staining prior to stimulation (day 0) and 14 days after stimulation (day 14) are shown. The HIV rev$_{67-75}$ peptide and C*05:01/HIV rev$_{67-75}$ multimer were used as controls. (D) Expansion of A*02:01/gp100$_{154-162}$ and A*02:01/tyrosinase$_{369-377}$ T cells in M87 TILs in an HLA-restricted peptide-specific manner. The M87 TILs were stimulated with A*02:01-artificial APCs pulsed with the indicated peptide. Data for the A*02:01/gp100$_{154-162}$ or A*02:01/tyrosinase$_{369-377}$ multimer staining prior to stimulation (day 0) and 14 days after stimulation (day 14) are shown. The HIV pol$_{476-484}$ peptide and A*02:01/HIV pol$_{476-484}$ multimer were employed as controls. (E) IFN-γ production by M87 TILs in an A*02:01/gp100$_{154-162}$-specific (left) or A*02:01/tyrosinase$_{369-377}$-specific (right) manner following peptide-specific stimulation. M87 TILs stimulated with A*02:01-artificial APCs pulsed with the gp100$_{154-162}$ or tyrosinase$_{369-377}$ peptide were used as responder cells. T2 cells pulsed with the indicated peptide were employed as stimulator cells. The HIV pol$_{476-484}$ peptide was used as a control. The percentage of multimer$^{+}$ cells in CD8$^{+}$ T cells is shown in (C) and (D). In (A), (B), and (E), the data shown represent the mean ± SD of experiments performed in triplicate. All the results are representative of at least two independent experiments. *p<0.05, **p<0.01, ***p<0.001. n.s., not statistically significant (two-tailed Welch's *t* tests).

The online version of this article includes the following figure supplement(s) for figure 2:

**Figure supplement 1.** Functional assessment of multimer-positive melanoma TILs.

antigens in the patient's own tumor tissues. Therefore, it is possible that the absence of a T cell response toward some shared antigens is due to the lack of their expression.

## Primary T cells transduced with TIL-derived TCR genes recognize tumor cells

The results of TCR gene therapy clinical trials demonstrate that the adoptive transfer of T cells transduced with high-affinity TCR genes can induce sustained clinical responses in cancer patients (*Feldman et al., 2015*; *Fesnak et al., 2016*; *Karpanen and Olweus, 2015*; *Morgan et al., 2006*; *Rapoport et al., 2015*; *Robbins et al., 2015*). Several tumor-reactive TCR genes have been cloned from melanoma TILs (*Feldman et al., 2015*; *Scheper et al., 2019*). Indeed, some of these TCRs have been tested in TCR gene therapy clinical trials and shown to induce clinically relevant responses (*Feldman et al., 2015*; *Karpanen and Olweus, 2015*). For the majority of tumor reactivities that we identified, multimer-positive antitumor T cells were collected and their TCR genes were molecularly cloned. All the HLA-B- and HLA-C-restricted T cell populations contained one pair of TCR genes

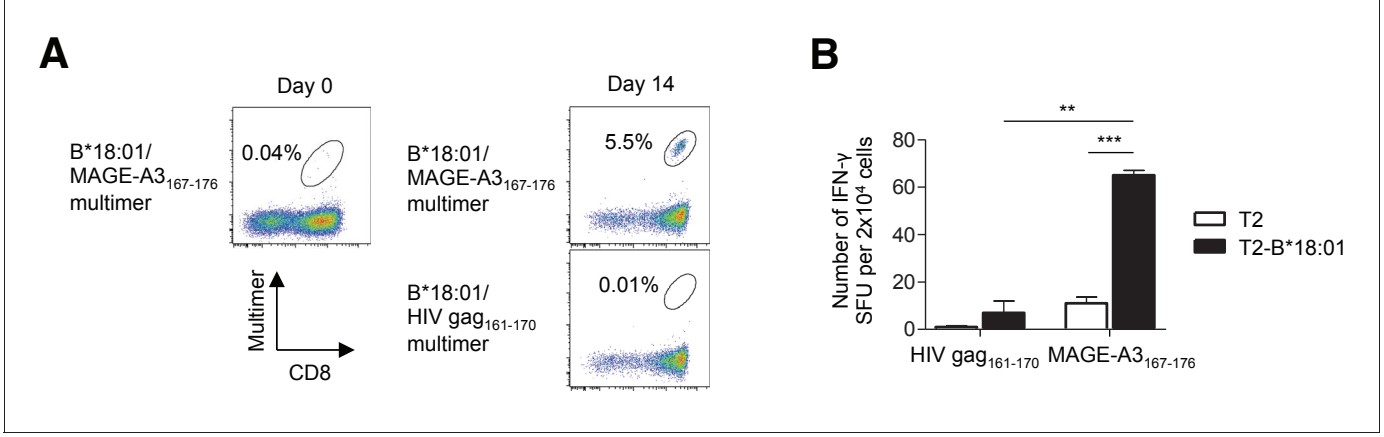

**Figure 3.** Identification of low-frequency antigen-specific T cells following peptide-specific expansion. CD8$^{+}$ T cells isolated from M40 TILs were stimulated with B*18:01-artificial APCs pulsed with 10 µg/ml MAGE-A3$_{167-176}$ peptide and cultured with 10 IU/ml IL-2, 10 ng/ml IL-15, and 30 ng/ml IL-21 for 14 days. (A) Data from staining with the indicated multimers before stimulation (day 0) and 14 days after stimulation (day 14) are shown. The B*18:01/HIV gag$_{161-170}$ multimer was used as a control. The percentage of multimer$^{+}$ cells in CD8$^{+}$ T cells is shown. (B) IFN-γ production by the M40 TILs in a B*18:01/MAGE-A3$_{167-176}$-specific manner following peptide-specific stimulation. CD8$^{+}$ T cells stimulated with B*18:01-artificial APCs pulsed with the MAGE-A3$_{167-176}$ peptide were employed as responder cells in IFN-γ ELISPOT analysis. T2 cells or T2 cells transduced with HLA-B*18:01 (T2-B*18:01) pulsed with MAGE-A3$_{167-176}$ or the HIV gag$_{161-170}$ control peptide were used as stimulator cells. The data shown represent the mean ± SD of experiments performed in triplicate. All the results are representative of at least two independent experiments. **p<0.01, ***p<0.001 (two-tailed Welch's *t* tests).

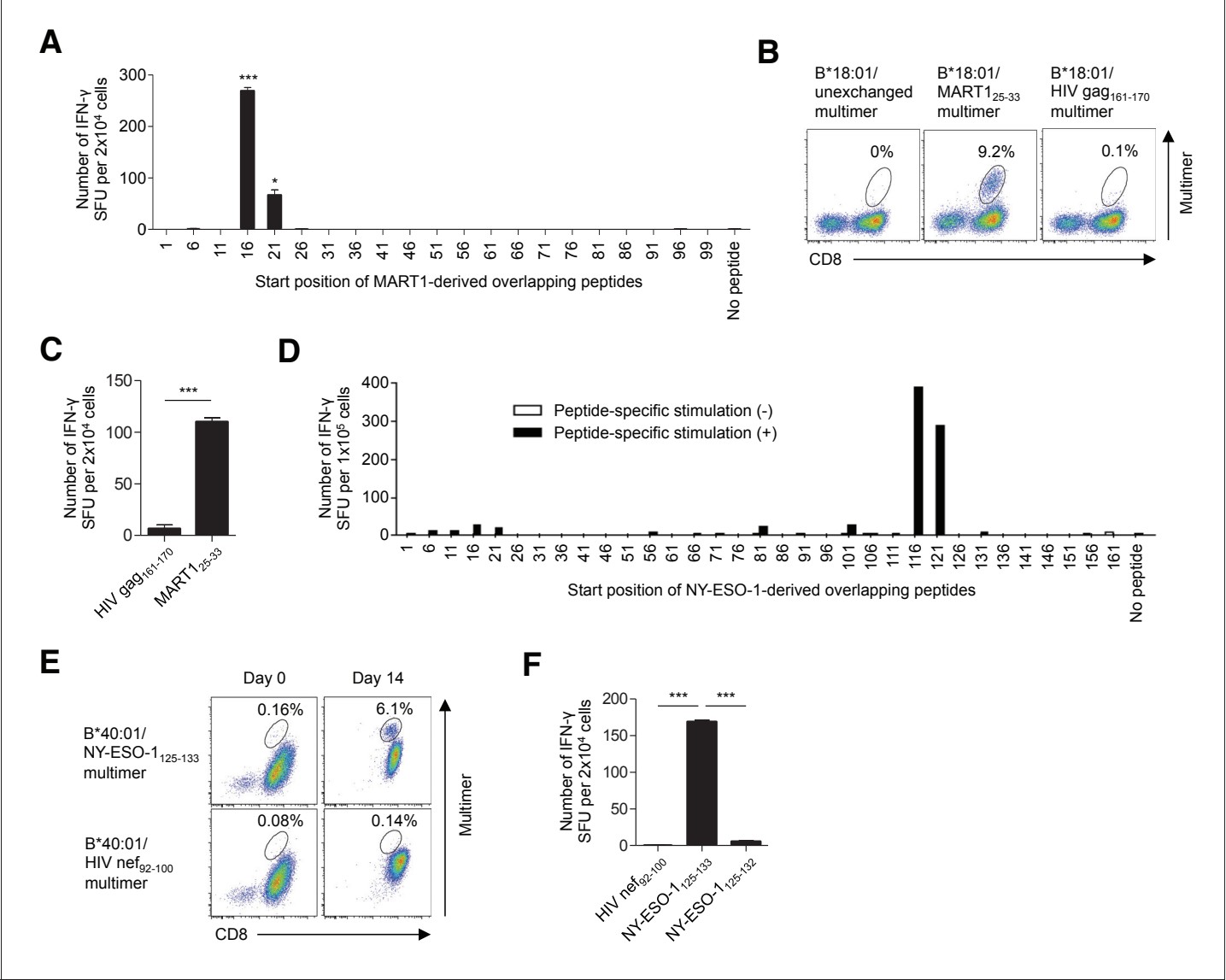

**Figure 4.** T cell epitope determination following stimulation with artificial APCs pulsed with overlapping peptides. (A–C) Identification of a new epitope B*18:01/MART1$_{25-33}$ in polyclonally expanded M87 TILs. (A) When stimulated with B*18:01-artificial APCs pulsed with MART1-derived overlapping peptides, the M87 TILs showed positive responses to two adjacent peptides with the shared sequence $_{21}$YTTAEEAAGIGILTV$_{35}$. See also **Supplementary file 2a**. (B) M87 TILs were stained with the indicated multimers. B*18:01/unexchanged and HIV gag$_{161-170}$ multimers were employed as controls. (C) IFN-γ production by M87 TILs in a B*18:01/MART1$_{25-33}$-specific manner. B*18:01-artificial APCs pulsed with MART1$_{25-33}$ or HIV gag$_{161-170}$ control peptide were employed as stimulator cells. (D–F) Identification of a new epitope, B*40:01/NY-ESO-1$_{125-133}$, recognized by polyclonally expanded M31 TILs. CD8$^+$ T cells isolated from M31 TILs were stimulated with B*40:01-artificial APCs pulsed with 10 µg/ml NY-ESO-1-derived overlapping peptides and cultured with 10 IU/ml IL-2, 10 ng/ml IL-15, and 30 ng/ml IL-21 for 14 days. (D) Following peptide-specific stimulation, the M31 TILs showed positive responses to two adjacent peptides with the shared sequence $_{121}$VLLKEFTVSGNILTI$_{135}$. See also **Supplementary file 2b**. (E) Data from B*40:01/NY-ESO-1$_{125-133}$ or control B*40:01/HIV nef$_{92-100}$ multimer staining before stimulation (day 0) and 14 days after stimulation (day 14) are shown. (F) IFN-γ production by M31 TILs in a B*40:01/NY-ESO-1$_{125-133}$-specific manner following peptide-specific stimulation. CD8$^+$ T cells stimulated with B*40:01-artificial APCs pulsed with the NY-ESO-1$_{125-133}$ peptide were employed as responder cells. B*40:01-artificial APCs pulsed with the indicated peptide were used as stimulator cells. The HIV nef$_{92-100}$ and NY-ESO-1$_{125-132}$ peptides were employed as controls. The percentage of multimer$^+$ cells in CD8$^+$ T cells is shown in (B) and (E). In (A), (C), and (F), the data shown represent the mean ± SD of experiments performed in triplicate. All the results are representative of at least two independent experiments. *p<0.05, ***p<0.001 (two-tailed Welch's *t* tests).

The online version of this article includes the following figure supplement(s) for figure 4:

**Figure supplement 1.** T cell epitope identification of in vitro-expanded TILs using overlapping peptides.

each. These results suggested that all the HLA-B- and HLA-C-restricted antigen-specific T cells found in this study were monoclonal (**Table 1**, **Figure 5—figure supplements 1** and **2**, **Supplementary file 3**). The antigen specificity and functional reactivity of the cloned TCRs were verified by multimer staining and ELISPOT assays of TCR-reconstituted T cells. For example, when reconstituted on primary T cells, B*18:01/MART1$_{25-33}$ TCR-transduced T cells were successfully stained with the cognate multimer (**Figure 5A**) and strongly reacted with the MART1$_{25-33}$ peptide presented by surface B*18:01 molecules (**Figure 5B**). Importantly, these cells were able to recognize B*18:01-matched and peptide-unpulsed tumor cells naturally expressing the MART1 gene (**Figure 5C**). Although both the Malme-3M and SK-MEL-28 melanoma cell lines are negative for B*18:01, they express the MART1 gene endogenously. When B*18:01 molecules were ectopically expressed, both melanoma cell lines were successfully recognized by B*18:01/MART1$_{25-33}$ TCR-transduced T cells. Moreover, A375 melanoma cells, which lack endogenous expression of both B*18:01 and MART1, became reactive to B*18:01/MART1$_{25-33}$ TCR-transduced T cells only when both the B*18:01 and full-length MART1 genes (but not either of the single genes) were transduced (**Figure 5C**, **Figure 5—figure supplements 6A** and **7B**).

Additionally, after transduction of the B*40:01/NY-ESO-1$_{125-133}$ TCR genes, primary T cells were successfully stained with a cognate multimer (**Figure 5D**) and strongly reacted with the NY-ESO-1$_{125-133}$ peptide presented by B*40:01 molecules on the cell surface (**Figure 5E**). Furthermore, these cells were able to recognize B*40:01-matched and peptide-unpulsed tumor cells endogenously expressing NY-ESO-1 (**Figure 5F**, **Figure 5—figure supplements 6A** and **7B**). These results clearly demonstrate that the B*18:01/MART1$_{25-33}$ and B*40:01/NY-ESO-1$_{125-133}$ TCR-transduced T cells were sufficiently avid to recognize tumor cells and that the cloned B*18:01/MART1$_{25-33}$ and B*40:01/NY-ESO-1$_{125-133}$ TCRs were both tumor-reactive. Using a similar strategy, we molecularly cloned the TCR genes of A*02:01/SSX2$_{41-49}$, A*24:02/gp100-intron4, B*07:02/NY-ESO-1$_{60-72}$, B*07:02/MAGE-A1$_{289-297}$, B*18:01/MAGE-A3$_{167-176}$, B*40:01/gp100$_{448-458}$, C*03:04/NY-ESO-1$_{92-100}$, C*05:01/tyrosinase$_{460-468}$, C*06:02/gp100$_{190-198}$, and C*07:01/gp100$_{479-487}$ and confirmed their tumor reactivities (**Table 1**, **Figure 5—figure supplements 2–7**, **Supplementary file 3**).

## Decoding the antigen specificity of polyclonally expanded TILs in melanoma

Using a library of pHLA class I multimers and artificial APCs, we were able to determine the specificity of 12.2 ± 7.3% (mean ± SD, max 25.9%, min 4.6%) of the CD8$^+$ TILs from eight melanoma patients toward 3.1 ± 2.0 (mean ± SD, max 7, min 1) previously known and novel peptides derived from shared antigens across HLA-A, B, and C. Notably, the maximum accumulative total percentage of multimer-positive CD8$^+$ T cells was 25.9% (M31 TILs) (**Figure 6**). At least one shared antigen-derived epitope was identified in all the TILs that were studied. One TIL sample (M87) possessed reactivity to as many as seven different antigens in the context of four different class I alleles. Interestingly, we observed that the M66 TIL sample reacted with three different A*02:01-restricted gp100-derived epitopes. Furthermore, M31 TILs exhibited reactivity to three different NY-ESO-1 peptides via three different class I alleles, and M87 TILs possessed reactivity to two different HLA-restricted epitopes each from gp100, tyrosinase, and MART1 (**Table 1**). These results underscore the established strong immunogenicity of these shared antigens.

## Discussion

HLA-restricted T cell responses toward immunogenic peptides, mutated or non-mutated, can induce antitumor effects in cancer patients (**Feldman et al., 2015**; **Rapoport et al., 2015**; **Robbins et al., 2015**; **Tran et al., 2016**). Since the HLA gene is the most polymorphic gene in the human genome (**Shao et al., 2018**), previous human T cell research has largely been limited to the few HLA alleles that are prevalent in a subset of ethnic groups. Therefore, a precise and comprehensive understanding of the antigen specificity for antitumor T cell responses remains lacking, including against non-mutated tumor antigens and infrequent HLA alleles. The sheer number of potential non-mutated antigens and the high polymorphism of HLA genes may have hampered comprehensive analysis of the specificity of antitumor T cell responses.

The adoptive transfer of tumor-reactive T cells, such as TILs or T cells transduced with high-affinity TCR genes, can induce sustained tumor regression in some cancer patients (**Feldman et al., 2015**;

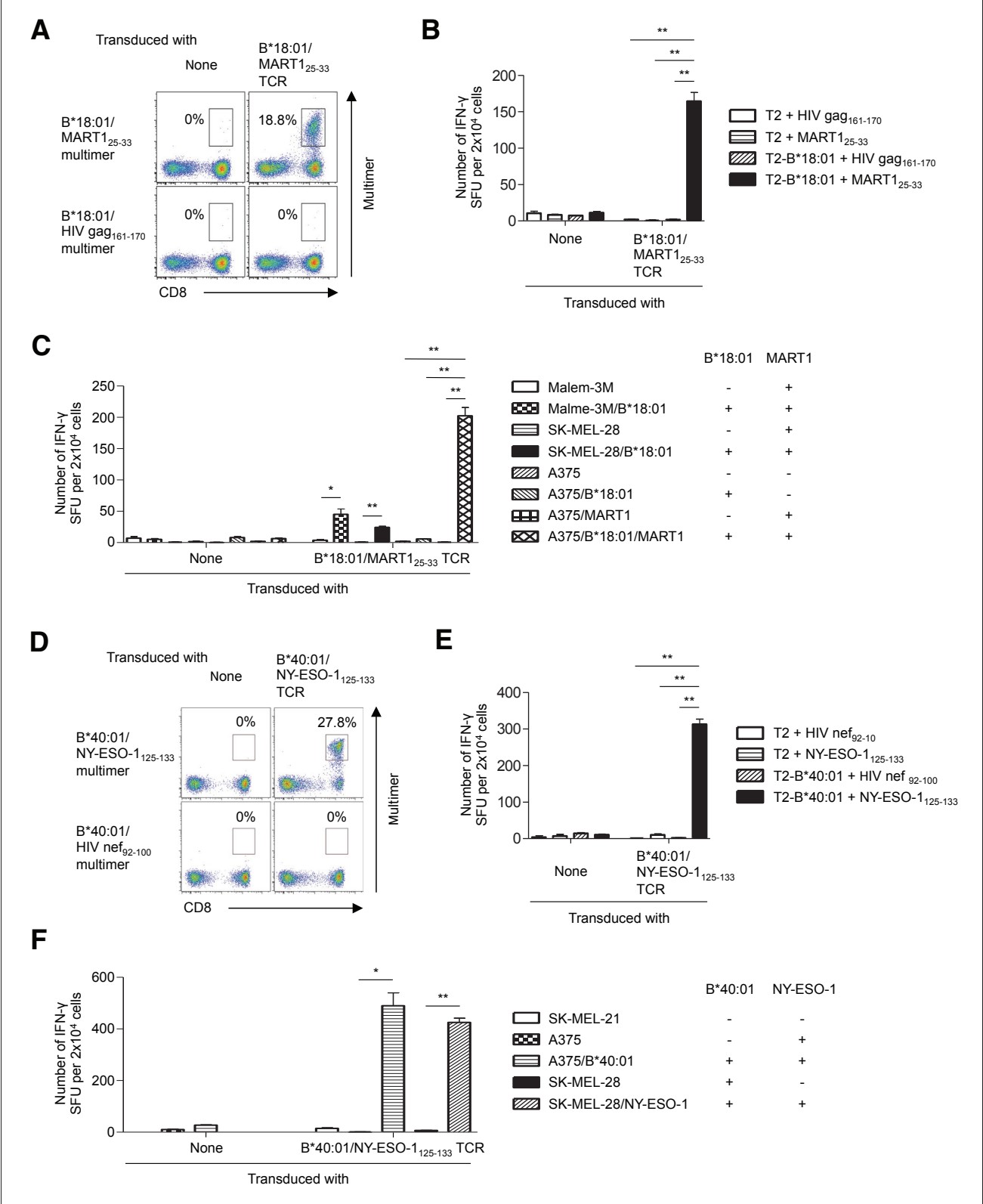

**Figure 5.** Tumor recognition by TIL-derived TCR genes. (**A–C**) Primary T cells transduced with the B*18:01/MART1$_{25-33}$ TCR recognized tumor cells. (**A**) B*18:01/MART1$_{25-33}$ TCR-transduced or untransduced T cells were stained with the B*18:01/MART1$_{25-33}$ or B*18:01/HIV gag$_{161-170}$ control multimer. (**B, C**) B*18:01/MART1$_{25-33}$ TCR-transduced or untransduced T cells were used as responder cells. (**B**) T2 or T2 cells transduced with HLA-B*18:01 (T2-B*18:01) pulsed with MART1$_{25-33}$ or HIV gag$_{161-170}$ control peptide were employed as stimulator cells. (**C**) Malme-3M, SK-MEL-28, and A375 cells that

*Figure 5 continued on next page*

*Figure 5 continued*

were either untransduced or transduced with HLA-B*18:01 and/or MART1 were used as stimulator cells. (D–F) Primary T cells transduced with the B*40:01/NY-ESO-1$_{125-133}$ TCR recognized tumor cells. (D) B*40:01/NY-ESO-1$_{125-133}$ TCR-transduced or untransduced T cells were stained with the B*40:01/NY-ESO-1$_{125-133}$ or B*40:01/HIV nef$_{92-100}$ control multimer. (E, F) B*40:01/NY-ESO-1$_{125-133}$ TCR-transduced or untransduced T cells were used as responder cells. (E) T2 or T2 cells transduced with HLA-B*40:01 (T2-B*40:01) pulsed with NY-ESO-1$_{125-133}$ or HIV nef$_{92-100}$ control peptide were employed as stimulator cells. (F) SK-MEL-21, A375, and SK-MEL-28 cells that were either untransduced or transduced with HLA-B*40:01 or NY-ESO-1 were used as stimulator cells. The percentage of multimer$^+$ CD8$^+$ T cells is shown in (A) and (D). In (B), (C), (E), and (F), the data shown represent the mean ± SD of experiments performed in triplicate. All the results are representative of at least two independent experiments. *p<0.05, **p<0.01 (two-tailed Welch's t tests).

The online version of this article includes the following figure supplement(s) for figure 5:

**Figure supplement 1.** Enrichment of A*02:01/SSX2$_{41-49}$ multimer-positive T cells.

**Figure supplement 2.** Confirmation of TCR specificity by reconstitution of TIL-derived TCR genes with a cognate multimer.

**Figure supplement 3.** Positive staining of human primary T cells transduced with TIL-derived TCR genes with a cognate multimer.

**Figure supplement 4.** Human primary T cells transduced with the TIL-derived TCR genes react strongly with the cognate peptide presented by the target class I molecule.

**Figure supplement 5.** Primary T cells transduced with TIL-derived TCR genes recognize tumor cells.

**Figure supplement 6.** Expression of target antigens derived from endogenous or transduced full-length genes.

**Figure supplement 7.** Expression of HLA class I derived from endogenous or transduced full-length genes.

*Rapoport et al., 2015*; *Robbins et al., 2015*; *Tran et al., 2016*). In this study, using a library of the paired pHLA multimers and artificial APCs for 25 different class I alleles and greater than 800 peptides, we systematically and comprehensively mapped shared antigenic epitopes recognized by TILs from eight melanoma patients for all their class I alleles. Furthermore, we isolated multiple TCR genes highly tumor-reactive to shared antigens from the TILs. In particular, NY-ESO-1 is one of the shared antigens that have been most promising and extensively studied, and many clinical trials using A*02:01/NY-ESO-1$_{157-165}$ TCR genes are ongoing. The use of these newly cloned tumor-reactive B*07:02, B*40:01, and C*03:04-restricted NY-ESO-1 TCR genes may widen the applicability of

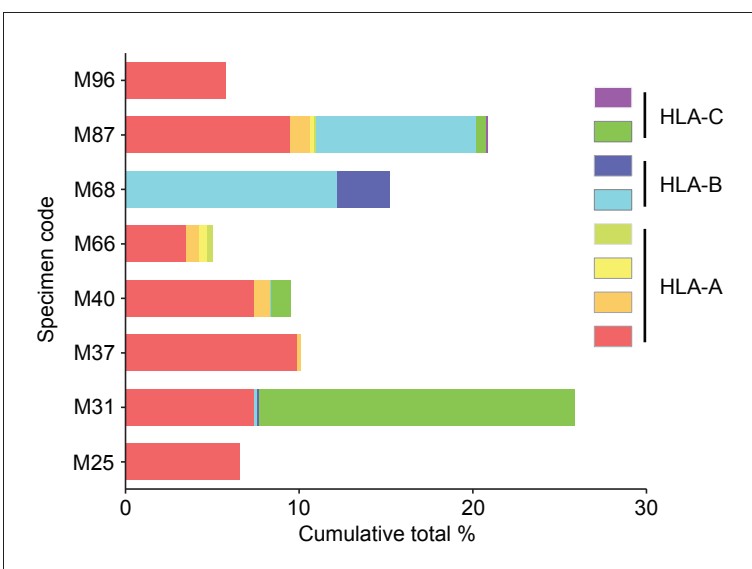

**Figure 6.** Cumulative percentages of antigen-specific T cells detected by multimer staining in polyclonally expanded TILs. The specificity of 12.2 ± 7.3% (mean ± SD, max 25.9%, min 4.6%) of CD8$^+$ T cells toward 3.1 ± 2.0 (mean ± SD, max 7, min 1) peptides derived from shared antigens was determined. The same colors do not represent the same alleles or antigens.

The online version of this article includes the following figure supplement(s) for figure 6:

**Figure supplement 1.** The efficiency of peptide exchange in soluble monomeric HLA class I$^{Q115E}$-K$^b$ monomers.

**Figure supplement 2.** Stability of the reagents after peptide loading and multimerization.

**Figure supplement 3.** Multiplex pHLA mutimer staining.

anti-NY-ESO-1 TCR gene therapy and immune mobilizing monoclonal TCRs against cancer (ImmTAC) therapy beyond HLA-A*02:01-positive cancer patients. ImmTAC therapy trials targeting A*02:01/gp100 are also ongoing in patients with metastatic uveal melanoma (*Liddy et al., 2012*). The newly cloned tumor-reactive gp100 TCRs restricted by four different class I alleles (A*24:02, B*40:01, C*06:02, and C*07:01) could substantially extend the applicability of ImmTAC therapy targeting gp100.

The strategy employed in this study has enabled us to decipher the antigen specificity of tumor-specific T cells for any given HLA class I allele, regardless of allele frequency (*Supplementary file 4*), and for a very large number of peptides. Although we studied T cell responses to greater than 800 peptides derived from more than 90 proteins across all the 25 class I alleles expressed by eight melanoma patients, our study is still limited. Additional comprehensive studies are required to elucidate the full spectrum of antitumor T cell response in TILs. Since in vitro expanded TILs were studied in the current study, further assessment of TIL directly ex vivo is needed in future studies. Our strategy has also allowed us to build a large database of class I-restricted peptides and cognate tumor-reactive TCR genes at an unprecedented scale. This database will facilitate individual examination of the personal immune response with precision as well as the identification and validation of biomarkers to aid in patient-based selection of a cancer immunotherapy regimen. Furthermore, this database will help the robust development of novel cancer vaccines, TCR gene therapies, and ImmTAC therapies for patients without being limited by HLA allele prevalence or tumor mutation burden.

## Materials and methods

### Key resources table

| Reagent type (species) or resource | Designation | Source or reference | Identifiers | Additional information |
|---|---|---|---|---|
| Cell line (*Homo-sapiens*) | K562 | ATCC | RRID:CVCL_0004 | |
| Cell line (*Homo-sapiens*) | T2 | ATCC | RRID:CVCL_2211 | |
| Cell line (*Homo-sapiens*) | A375 | ATCC | RRID:CVCL_0132 | |
| Cell line (*Homo-sapiens*) | Malme-3M | ATCC | RRID:CVCL_1438 | |
| Cell line (*Homo-sapiens*) | SK-MEL-28 | ATCC | RRID:CVCL_0526 | |
| Cell line (*Homo-sapiens*) | LM-MEL-53 | ATCC | RRID:CVCL_UC56 | |
| Cell line (*Homo-sapiens*) | HEK293T | ATCC | RRIID:CVCL_0063 | |
| Cell line (*Homo-sapiens*) | MCF7 | ATCC | RRID:CVCL_UC56 | |
| Cell line (*Homo-sapiens*) | ACHN | ATCC | RRID:CVCL_1067 | |
| Cell line (*Homo-sapiens*) | SK-MEL-21 | Memorial Sloan Kettering Cancer Center | RRID:CVCL_3877 | |
| Cell line (*Homo-sapiens*) | SK-MEL-37 | Memorial Sloan Kettering Cancer Center | RRID:CVCL_3878 | |
| Cell line (*Homo-sapiens*) | Me275 | Ludwig Institute for Cancer Research | RRID:CVCL_S597 | |
| Cell line (*Homo-sapiens*) | Jurkat 76 | Leiden University Medical Center | | *Heemskerk et al. (2003)* |
| Antibody | Mouse anti-CD8-PC5 | Beckman Coulter | Cat.#: IM2638U | Clone B9.11 |
| Antibody | Mouse CD8-PB | Beckman Coulter | Cat.#: A82791 | Clone B9.11 |

*Continued on next page*

*Continued*

| Reagent type (species) or resource | Designation | Source or reference | Identifiers | Additional information |
|---|---|---|---|---|
| Antibody | Mouse anti-NGFR-FITC | Biolegend | Cat.#: MAB5598 | Clone ME20.4 |
| Antibody | Mouse anti-CD3-APC/Cy7 | Biolegend | Cat.#: 300426 | Clone UCHT1 |
| Antibody | Mouse anti-HLA-A2-FITC | Biolegend | Cat.#: 343304 | Clone BB7.2 |
| Antibody | Mouse anti-Melan-A | Santa Cruz Biotechnology | Cat.#: sc-20032 | Clone A103 |
| Antibody | Rabbit anti-NY-ESO-1-PE | Cell Signaling Technology | Cat.#: 65699 | Clone D1Q2U |
| Antibody | Mouse anti-PMEL | LifeSpan Biosciences | Cat.#: LS-C114528 | Clone 7E3 |
| Antibody | Mouse anti-MAGE-A1 | LifeSpan Biosciences | Cat.#: LS-C391567 | Clone MA454 |
| Antibody | Rabbit anti-Tyrosinase | Abcam | Cat.#: ab61294 | Clone ERP10141 |
| Antibody | Rabbit anti-SSX2 | Thermo Fisher Scientific | Cat.#: PA5-80074 | Polyclonal |
| Antibody | Rabbit anti-MAGE-A3 | LifeSpan Biosciences | Cat.#: LS-C99934 | Polyclonal |
| Antibody | Mouse anti-β-actin | Santa Cruz Biotechnology | Cat.#: sc-47778 | Clone C4 |
| Antibody | Mouse anti-His tag-PE | Abcam | Cat.#: ab72467 | Clone AD1.1.10 |
| Antibody | Mouse anti-His tag-APC | Abcam | Cat.#: ab72579 | Clone AD1.1.10 |
| Commercial assay or kit | LIVE/DEAD Fixable Aqua Dead Cell Stain kit | Thermo Fisher Scientific | at.#: L34957 | |
| Commercial assay or kit | PureLink Genomic DNA Mini Kit | Thermo Fisher Scientific | Cat.#: K182000 | |
| Commercial assay or kit | Cytofix/Cytoperm kit | BD Biosciences | Cat.#: BD 554714 | |
| Commercial assay or kit | CD8+ T Cell Isolation Kit | Miltenyi Biotec | Cat.#: 130-096-495 | |
| Commercial assay or kit | Pan T Cell Isolation Kit | Miltenyi Biotec | Cat.#: 130-096-535 | |
| Software, algorithm | Cancer Antigenic Peptide Database | Cancer Research Institute | https://caped.icp.ucl.ac.be | |
| Software, algorithm | NetMHC 4.0 | *Andreatta and Nielsen (2016)* | http://www.cbs.dtu.dk/services/NetMHC/ | |

## Cell samples

Peripheral blood samples were obtained from healthy donors after Institutional Review Board approval. Mononuclear cells were obtained via density gradient centrifugation (Ficoll-Paque PLUS; GE Healthcare). K562 is an erythroleukemic cell line with defective HLA expression. T2 is an HLA-A*02:01+ T cell leukemia/B-LCL hybrid cell line. Jurkat 76 is a T cell leukemic cell line lacking TCR and CD8 expression (a gift from Dr. M. Heemskerk, Leiden University Medical Center, Leiden, the Netherlands) (*Heemskerk et al., 2003*). A375, Malme-3M, SK-MEL-21, SK-MEL-28, SK-MEL-37, Me275, and LM-MEL-53 are melanoma cell lines. The HEK293T, MCF7 and melanoma cell lines, except for Malme-3M and LM-MEL-53, were grown in DMEM supplemented with 10% FBS and 50 μg/ml gentamicin (Thermo Fisher Scientific). Malme-3M was cultured in IMDM supplemented with 20% FBS and 50 μg/ml gentamicin. The K562, T2, Jurkat 76, and LM-MEL-53 cell lines were grown in RPMI 1640 supplemented with 10% FBS and 50 μg/ml gentamicin. ACHN was cultured in EMEM supplemented with 10% FBS and 50 μg/ml gentamicin. The K562, T2, A375, Malme-3M, SK-MEL-28, LM-MEL-53, HEK293T, MCF7, and ACHN cells were obtained from the American Type Culture Collection (ATCC, Manassas, VA). The SK-MEL-21 and SK-MEL-37 cells were obtained from Memorial Sloan Kettering Cancer Center (New York, NY). The Me275 cells were obtained from Ludwig Institute for Cancer Research (New York, NY). All cell lines were routinely checked for the presence of mycoplasma contamination using PCR-based technology. TILs isolated from eight metastatic

melanoma patients were grown in vitro as reported previously (*Nguyen et al., 2019*). High-resolution HLA DNA typing (American Red Cross) was performed for all TIL samples. Melanoma specimens were obtained from UHN Biospecimen Program. This study was conducted in accordance with the Helsinki Declaration and approved by the Research Ethics Board of the University Health Network, Toronto, Canada. Written informed consent was obtained from all healthy donors who provided peripheral blood samples.

## Peptides

Synthetic peptides were purchased from Genscript (Piscataway, NJ) and dissolved at 50 mg/ml in DMSO. The purity of the vast majority of the peptides exceeded 85%. The peptide sequences are shown in *Supplementary files 1* and *2.*

## Genes

All the HLA class I genes except for HLA-A*02:01 were fused with a truncated version of the human nerve growth factor receptor (ΔNGFR) via the internal ribosome entry site (*Renaud-Gabardos et al., 2015*). ΔNGFR-transduced cells were isolated using anti-NGFR mAb. The full-length MART1, NY-ESO-1, SSX2, and tyrosinase genes were cloned from Malme-3M, Me275, SK-MEL-37, and SK-MEL-28 cells via RT-PCR according to their published sequences, respectively. The full-length gp100, MAGE-A1, and MAGE-A3 genes were purchased from Dharmacon (Lafayette, CO). Genomic DNA of gp100 was isolated using PureLink Genomic DNA Mini Kit (Thermo Fisher Scientific). TCR genes were cloned via 5'-rapid amplification of cDNA ends (RACE) PCR as previously described (*Nakatsugawa et al., 2016*). The 5'-RACE PCR products were cloned into a retrovirus vector and sequenced. All genes were cloned into the pMX retrovirus vector and transduced using the 293GPG cell-based retrovirus system (*Ory et al., 1996*).

## Transfectants

Jurkat 76/CD8 cells were transduced with individual TCRα and TCRβ genes as reported previously (*Ochi et al., 2015*). The Jurkat 76/CD8-derived TCR transfectants were purified (>95% purity) using CD3 Microbeads (Miltenyi Biotec). The K562-based artificial APCs individually expressing various HLA class I genes as a single HLA allele in conjunction with CD80 and CD83 have been reported previously (*Butler and Hirano, 2014*). PG13-derived retrovirus supernatants were used to transduce TCR genes into human primary T cells. TransIT293 (Mirus Bio) was used to transfect TCR genes into the 293GPG cell line. SSX2⁻ SK-MEL-21 and SK-MEL-28 cells were retrovirally transduced with the full-length SSX2 gene to generate SK-MEL-21/SSX2 and SK-MEL-28/SSX2 cells, respectively. Gp100⁻ A375 cells were retrovirally transduced with exons 1, 2, and 3 and intron 4 of the gp100 gene to generate A375/gp100-intron4 cells as reported previously (*Robbins et al., 1997*). Gp100⁻ SK-MEL-37, ACHN, and A375 cells were retrovirally transduced with the full-length gp100 gene to generate SK-MEL-37/gp100, ACHN/gp100, and A375/gp100 cells, respectively. MART1⁻ A375 cells were retrovirally transduced with the full-length MART1 gene to generate A375/MART1 cells. NY-ESO-1⁻ SK-MEL-21 and SK-MEL-28 cells were retrovirally transduced with the full-length NY-ESO-1 gene to generate SK-MEL-21/NY-ESO-1 and SK-MEL-28/NY-ESO-1 cells, respectively. MAGE-A1⁻ SK-MEL-21 cells were retrovirally transduced with the full-length MAGE-A1 gene to generate SK-MEL-21/MAGE-A1 cells. MAGE-A3⁻ HEK293T cells were retrovirally transduced with the full-length MAGE-A3 gene to generate HEK293T/MAGE-A3 cells. Tyrosinase⁻ MCF7 cells were retrovirally transduced with the full-length tyrosinase gene to generate MCF7/tyrosinase cells. The expression of transduced MART1, NY-ESO-1, gp100, MAGE-A1, and tyrosinase was evaluated by flow cytometry after staining with an anti-MART1 mAb (clone A103; Santa Cruz Biotechnology), anti-NY-ESO-1 mAb (clone D1Q2U; Cell Signaling Technology), anti-gp100 mAb (clone 7E3; LifeSpan Biosciences), anti-MAGE-A1 mAb (clone MA454; LifeSpan Biosciences), and anti-tyrosinase mAb (clone ERP10141; Abcam), respectively. The expression of SSX2 and MAGE-A3 in the transduced cells was evaluated by Western blot analysis with an anti-SSX2 pAb (Thermo Fisher Scientific) and anti-MAGE-A3 pAb (LifeSpan Biosciences), respectively.

HLA-A*02:01⁻ SK-MEL-28 cells were retrovirally transduced with the HLA-A*02:01 gene to generate SK-MEL-28/A*02:01. HLA-A*24:02⁻ Malme-3M, SK-MEL-28, and A375 cells were retrovirally transduced with HLA-A*24:02 to generate Malme-3M/A*24:02, SK-MEL-28/A*24:02, and A375/

A*24:02 cells, respectively. Similarly, A375/B*07:02, SK-MEL-37/B*07:02, Me275/B*07:02 cells, Malme-3M/B*18:01, SK-MEL-28/B*18:01, A375/B*18:01, HEK293T/B*18:01, A375/B*40:01, SK-MEL-37/B*40:01, A375/C*03:04, SK-MEL-37/C*03:04, Malme-3M/C*05:01, Me275/C*05:01, Malme-3M/C*06:02, SK-MEL28/C*06:02, and A375/C*07:01 cells were generated using a retrovirus system. All the class I genes except for A*02:01, were tagged with the ΔNGFR gene as described above, and the ΔNGFR⁺ cells were purified (>95% purity) and used in subsequent experiments. The ΔNGFR gene alone was retrovirally transduced as a control.

## Flow cytometry and cell sorting

Cell surface molecules were stained with a PC5- or Pacific Blue-conjugated anti-CD8 mAb (clone B9.11; Beckman Coulter), FITC-conjugated anti-NGFR mAb (clone ME20.4; Biolegend), APC/Cy7-conjugated anti-CD3 mAb (clone UCHT1; Biolegend), and FITC-conjugated anti-HLA-A2 mAb (clone BB7.2; Biolegend). Dead cells were discriminated with the LIVE/DEAD Fixable Aqua Dead Cell Stain kit (Thermo Fisher Scientific). For intracellular staining, cells were fixed and permeabilized by using a Cytofix/Cytoperm kit (BD Biosciences). Stained cells were analyzed with flow cytometry (BD Biosciences), and data analysis was performed using FlowJo (Tree Star). Cell sorting was conducted using a FACS Aria II (BD Biosciences).

## Cytokine ELISPOT analysis

IFN-γ ELISPOT analysis was conducted as described previously (*Kagoya et al., 2018*). PVDF plates (Millipore, Bedford, MA) were coated with the capture mAb (clone 1-D1K; MABTECH, Mariemont, OH), and T cells were incubated with $2 \times 10^4$ target cells per well in the presence or absence of a peptide for 20–24 hr at 37°C. The plates were subsequently washed and incubated with a biotin-conjugated detection mAb (clone 7-B6-1; MABTECH). HRP-conjugated SA (Jackson ImmunoResearch) was then added, and IFN-γ spots were developed. The reaction was stopped by rinsing thoroughly with cold tap water. ELISPOT plates were scanned and counted using an ImmunoSpot plate reader and ImmunoSpot version 5.0 software (Cellular Technology Limited, Shaker Heights, OH).

## Immunoblotting

Equal amounts of proteins were separated on 8% gels by SDS-PAGE and transferred to Immobilon-P PVDF membranes (Millipore). The membranes were probed with the primary antibodies at 4°C overnight. The membranes were then washed and incubated with HRP-conjugated anti-mouse IgG (Promega) or anti-rabbit IgG (Santa Cruz Biotechnology) secondary antibody at room temperature for 1 hr. The following antibodies were used: anti-SSX2 pAb (Thermo Fisher Scientific), anti-MAGE-A3 pAb (LifeSpan Biosciences), and anti–β-actin antibody (Santa Cruz Biotechnology). The signal was detected by Amersham ECL Prime Western Blotting Detection Reagent (GE Healthcare).

## TIL culturing and rapid expansion protocol (REP)

The TIL expansion procedure was performed as previously published (*Nguyen et al., 2019*). Briefly, melanoma tissue was processed by cutting into ~1 mm³ fragments. Tissue fragments were either plated directly into 24-well plates or enzymatically dissociated in IMDM containing collagenase (Sigma) and Pulmozyme (Roche) and then plated in 24-well plates. Cells were cultured in complete medium (as previously described) and 6,000 IU/mL IL-2 and expanded for approximately 4 weeks prior to cryopreservation.

For the REP, TILs were thawed, rested, and seeded in T175 flasks with 30 ng/mL OKT3 (Miltenyi Biotec), irradiated (50 Gy) allogeneic PBMC feeder cells (1:200 TIL:PBMC), and 600 IU/mL IL-2 in '50/50' media containing 50% complete medium prepared using human serum AB⁺ (Gemini Bio Products) and 50% AIM V media (Gibco). TILs were harvested on day 14 of the REP and cryopreserved before analysis.

## Expansion of CD8⁺ TILs in an HLA-restricted peptide-specific manner

CD8⁺ TILs were purified through negative magnetic selection using a CD8⁺ T Cell Isolation Kit (Miltenyi Biotec). HLA class I-matched artificial APCs were pulsed with 10 μg/ml class I-restricted peptides of interest for 6 hr. The artificial APCs were then irradiated at 200 Gy, washed, and added to the TILs at an effector to target (E:T) ratio of 20:1. After forty-eight hours, 10 IU/ml IL-2 (Novartis),

10 ng/ml IL-15 (Peprotech), and 30 ng/ml IL-21 (Peprotech) were added to the cultures every three days.

## Expansion of primary CD8$^+$ T cells transduced with cloned TCRs

CD3$^+$ T cells were purified through negative magnetic selection using a Pan T Cell Isolation Kit (Miltenyi Biotec). Purified T cells were stimulated with artificial APC/mOKT3 irradiated with 200 Gy at an E:T ratio of 20:1. Starting on the next day, activated T cells were retrovirally transduced with the cloned TCR genes via centrifugation for 1 hr at 1,000 g at 32°C for three consecutive days. On the following day, 100 IU/ml IL-2 and 10 ng/ml IL-15 were added to the TCR-transduced T cells. The culture medium was replenished every 2–3 days.

## Production of human cell-based pHLA multimers

The affinity-matured HLA class I gene was engineered to carry a Glu (E) residue in lieu of the Gln (Q) residue at position 115 of the α2 domain and a mouse K$^b$ gene-derived α3 domain instead of the HLA class I α3 domain (*Wooldridge et al., 2007*). By fusing the extracellular domain of the affinity-matured HLA class I gene with a Gly-Ser (GS) flexible linker followed by a 6x His tag, we generated the soluble HLA class I$^{Q115E}$-K$^b$ gene. HEK293T cells were individually transduced with various soluble HLA class I$^{Q115E}$-K$^b$ genes using the 293GPG cell-based retrovirus system (*Ory et al., 1996*). Stable HEK293T cells expressing soluble affinity-matured class I$^{Q115E}$-K$^b$ gene were grown until confluent, and the medium was then changed. Forty-eight hours later, the conditioned medium was harvested and immediately used or frozen until use. The soluble HLA class I$^{Q115E}$-K$^b$-containing supernatant produced by the HEK293T transfectants was mixed with 100–1000 µg/ml of class I-restricted peptide of interest overnight at 37°C for in vitro peptide exchange. Soluble monomeric class I$^{Q115E}$-K$^b$ loaded with the peptide was multimerized using an anti-His mAb (clone AD1.1.10; Abcam) conjugated to a fluorochrome such as phycoerythrin (PE) or allophycocyanin (APC) at a 2:1 molar ratio for 2 hr at room temperature or overnight at 4°C. The concentration of functional soluble HLA class I$^{Q115E}$-K$^b$ molecules was measured by specific enzyme-linked immunosorbent assay (ELISA) using an anti-pan class I mAb (clone W6/32; in-house) and an anti-His tag biotinylated mAb (clone AD1.1.10; R and D systems) as capture and detection Abs, respectively.

## Measurement of peptide exchange efficiency using ELISA

The efficiency of peptide exchange in soluble class I$^{Q115E}$-K$^b$ monomers was assessed using a competition binding assay and ELISA. The monomer was loaded with 100 µg/ml biotinylated peptide and incubated overnight at 37°C. Biotinylated peptide-monomer was purified and exchanged with PBS using Amicon Ultra filters (molecular weight cut-off (MWCO) 10 kDa) (Millipore Sigma, Burlington, MA) and mixed with 1 mg/ml HLA-restricted peptide of interest or an equivalent volume of DMSO followed by overnight incubation at 37°C. ELISA plates were coated with anti-pan class I mAb (clone W6/32) at 10 µg/mL in PBS overnight at 4°C. The plates were washed and blocked with 10% nonfat dry milk in PBS for 30 min at room temperature. Peptide-monomer was added and incubated for 2 hr at room temperature. After washing, the plates were incubated with streptavidin-conjugated alkaline phosphatase for 30 min at room temperature. Finally, the plates were washed and incubated with p-nitrophenyl phosphate (PNPP) substrate (Pierce, Rockford, IL) at room temperature. The reaction was terminated by adding 1 mol/L NaOH. The optic density (OD) (405 nm) was read (Spectramax 190 Microplate Reader; Molecular Devices, Sunnyvale, CA). The OD values from the control wells containing nonbiotinylated peptide were subtracted from the OD values in test wells containing biotinylated peptide. The efficiency of peptide exchange for each monomer was calculated as follows: Peptide exchange efficiency = [1 - (OD value with peptide/OD value with DMSO)] x 100. Every sample was assayed in triplicate wells. The biotinylated peptides used were HLA-A*02:01-restricted telomerase$_{540-548}$ ILAK(-biotin)FLHWL, B*07:02-restricted MiHAg SMCY$_{1041-1051}$ SPSVDK(-biotin)ARAEL, and C*07:02-restricted adenovirus B$_{585-593}$ FRK(-biotin)DVNMVL. The efficiencies of peptide exchange in the A*02:01, B*07:02, and C*07:02 monomers were shown in *Figure 6—figure supplement 1*.

## pHLA multimer staining

T cells ($1 \times 10^5$) were incubated for 30 min at 37°C in the presence of 50 nM dasatinib (LC laboratories). The cells were then washed and incubated with 5–10 µg/ml of PE-conjugated multimer for 30 min at room temperature, and R-phycoerythrin-conjugated AffiniPure Fab fragment goat anti-mouse IgG1 antibody (Jackson ImmunoResearch Laboratories) was added for 15 min at 4°C. Next, the cells were washed three times and costained with an anti-CD8 mAb for 15 min at 4°C. Dead cells were finally discriminated using the LIVE/DEAD Fixable Dead Cell Stain kit. For multiplex staining, after incubation with PE-conjugated multimer, the cells were washed three times and then incubated with 5–10 µg/ml of APC-conjugated multimer for 30 min at room temperature followed by costaining with an anti-CD8 mAb. The stability and multiplexing ability of the reagents were confirmed (*Figure 6—figure supplements 2* and *3*).

## Statistical analysis

Statistical analysis was performed using GraphPad Prism 5.0. To determine whether two groups were significantly different for a given variable, we conducted an analysis using Welch's t test (two-sided). P values < 0.05 were considered significant.

## Acknowledgements

We thank the Princess Margaret Melanoma/Skin Oncology Disease Site, BioBank, and Cell Production Team for collecting melanoma samples and generating TILs. This work was supported by an Ontario Institute for Cancer Research Clinical Investigator Award IA-039 (NH); the Terry Fox immunotherapy NeTwork (iTNT) Program (PSO, NH); the Princess Margaret Cancer Centre Innovation Accelerator Fund (NH); the Princess Margaret Cancer Foundation (NH, MOB); the Mitacs Internship (KM); a Japan Society for the Promotion of Science Postdoctoral Fellowship for Overseas Researchers and a Guglietti Fellowship (YK); the Province of Ontario (TG, MA); and the Natural Sciences and Engineering Research Council of Canada Postgraduate Scholarship (TG). The funders had no role in study design, data collection and analysis, decision to publish, or preparation of the manuscript.

## Additional information

### Competing interests

Kenji Murata: The University Health Network has filed patent application related to this study on which Kenji Murata is named as an inventor (US16/095,913, US62/813,639, US62/813,642, US62/813,644, US62/813,645, US62/813,647, US62/813,650, US62/813,651, and US62/823,487). Munehide Nakatsugawa: The University Health Network has filed patent application related to this study on which Munehide Nakatsugawa is named as an inventor (US16/095,913). Muhammed A Rahman: The University Health Network has filed patent application related to this study on which Muhammed A. Rahman is named as an inventor (US16/095,913). Kayoko Saso: The University Health Network has filed patent application related to this study on which Kayoko Saso is named as an inventor (US62/813,639, US62/813,642, US62/813,644, US62/813,645, US62/813,647, US62/813,650, US62/813,651, and US62/823,487). Marcus O Butler: Marcus O. Butler has served on advisory boards for Merck, BMS, Novartis, GSK, Immunocore, immunovaccine, Sanofi, and EMD Serono and received research funding for investigator initiated clinical trials from Merck and Takara Bio. Naoto Hirano: Naoto Hirano reports receiving a research grant from and is a consultant for Takara Bio and Otsuka Pharmaceutical and serving on an advisory board for F. Hoffmann-La Roche. The University Health Network has filed patent application related to this study on which Naoto Hirano is named as an inventor (US16/095,913, US62/813,639, US62/813,642, US62/813,644, US62/813,645, US62/813,647, US62/813,650, US62/813,651, and US62/823,487). The other authors declare that no competing interests exist.

## Funding

| Funder | Grant reference number | Author |
| --- | --- | --- |
| Ontario Institute for Cancer Research | Clinical Investigator Award, IA-039 | Naoto Hirano |
| Terry Fox Research Institute | Immunotherapy NeTwork (iTNT) Program | Pamela S Ohashi Naoto Hirano |
| Princess Margaret Cancer Centre | Innovation Accelerator Fund | Naoto Hirano |
| Princess Margaret Cancer Foundation | | Marcus O Butler Naoto Hirano |
| Mitacs | Intership | Kenji Murata |
| Japan Society for the Promotion of Science | Postdoctoral Fellowship for Overseas Researchers | Yuki Kagoya |
| Province of Ontario | | Tingxi Guo Mark Anczurowski |
| Natural Sciences and Engineering Research Council of Canada | Postgraduate Scholarship | Tingxi Guo |
| Guglietti Fellowship | | Yuki Kagoya |

The funders had no role in study design, data collection and interpretation, or the decision to submit the work for publication.

## Author contributions

Kenji Murata, Conceptualization, Investigation, Writing - original draft, Writing - review and editing; Munehide Nakatsugawa, Conceptualization, Investigation; Muhammed A Rahman, Linh T Nguyen, Douglas G Millar, Kenji Sugata, Hiroshi Saijo, Yukiko Matsunaga, Yuki Kagoya, Tingxi Guo, Mark Anczurowski, Chung-Hsi Wang, Brian D Burt, Dalam Ly, Kayoko Saso, Investigation; David T Mulder, Trevor J Pugh, Methodology; Alexandra Easson, David P Goldstein, Michael Reedijk, Danny Ghazarian, Tak W Mak, Resources; Marcus O Butler, Pamela S Ohashi, Resources, Writing - review and editing; Naoto Hirano, Conceptualization, Supervision, Writing - original draft, Project administration, Writing - review and editing

## Author ORCIDs

Kenji Murata https://orcid.org/0000-0002-2613-5144
Naoto Hirano https://orcid.org/0000-0001-9070-4754

## Ethics

Human subjects: This study was conducted in accordance with the Helsinki Declaration and approved by the Research Ethics Board of the University Health Network, Toronto, Canada (UHN REB# 11-0343, 11-0348). Written informed consent was obtained from all healthy donors who provided peripheral blood samples.

## Decision letter and Author response

Decision letter https://doi.org/10.7554/eLife.53244.sa1
Author response https://doi.org/10.7554/eLife.53244.sa2

# Additional files

## Supplementary files

• Supplementary file 1. List of previously known and newly predicted HLA-bound peptides. (a) Peptides used for initial A*01:01 multimer staining. (b) Peptides used for initial A*02:01 multimer staining and measurement of peptide exchange efficiency. (c) Peptides used for initial A*24:02 multimer staining. (d) Peptides used for initial B*07:02 multimer staining and measurement of peptide

exchange efficiency. (e) Peptides used for initial B*08:01 multimer staining. (f) Peptides used for initial C*05:01 multimer staining. (g) Peptides used for initial C*07:01 multimer staining. (h) Peptides used for initial C*07:02 multimer staining and measurement of peptide exchange efficiency. (i) Other peptides used for initial pHLA multimer staining.

• Supplementary file 2. List of overlapping peptides covering whole proteins. (a) MART1-derived overlapping peptides. (b) NY-ESO-1-derived overlapping peptides. (c) SSX2-derived overlapping peptides. (d) gp100-derived overlapping peptides. (e) MAGE-A1-derived overlapping peptides. (f) Survivin-derived overlapping peptides.

• Supplementary file 3. CDR3 sequences of TCR genes isolated from multimer-positive TILs.

• Supplementary file 4. Twenty-five class I alleles for which paired multimers and artificial APCs were generated.

• Transparent reporting form

### Data availability

All data generated or analysed during this study are included in the manuscript and supporting files.

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
