## [Decision Letter]

**Decision letter after peer review:**

Thank you for submitting your article "Landscape mapping of shared antigenic epitopes and their cognate TCRs of tumor-infiltrating T lymphocytes in melanoma" for consideration by *eLife*. Your article has been reviewed by Jeffrey Settleman as the Senior Editor, a Reviewing Editor, and three reviewers. The following individuals involved in review of your submission have agreed to reveal their identity: Ravi Sachidanandam (Reviewer #1); Kunle Odunsi (Reviewer #3).

The reviewers have discussed the reviews with one another and the Reviewing Editor has drafted this decision to help you prepare a revised submission.

Summary:

The manuscript by Murata, Nakatsugawa et al., describes extensive testing of melanoma infiltrating T-cell reactivity against a large panel of shared tumor antigen-derived epitopes to comprehensively analyze tumor antigen-specific T-cell response in the tumor microenvironment. More than 800 synthetic peptides that were previously reported or expected to bind on 25 different HLA types were tested. To enable cost- and labor-effective testing, the group developed a novel peptide-exchangeable pHLA class I multimer technology. Low-frequency responses by multimer-staining were validated by additional experiments such as IFN-γ ELISPOT assays and peptide-specific T-cell expansion using their unique artificial antigen-presenting cells (aAPCs). Finally, the group cloned T-cell receptor (TCR) genes from multimer-reactive T cells and demonstrated tumor-recognition by TCR gene-engineered T cells. The strategy can be useful in the development of personalized cancer vaccines or adoptive T-cell therapy using TCR gene-engineered T cells. This is a very intensive and meticulous effort in understanding T-cell reactivity in the tumor microenvironment, and the authors are to be commended for their approach.

Essential revisions:

Figure 1 which shows the results from a screening experiment – in total this figure shows 200 FACS plots. The presentation of the data needs to be reconsidered to emphasize the key conclusions.

Sequencing the TCR repertoires to determine the actual clonality would have been very useful. But it does not seem to have been successful in most cases. Was it because the expanded clones were still polyclonal? A related question, how often are they polyclonal? This could be an interesting question to address.

What about the obvious control, the circulating T cells from the patients, to see if they have a different characteristic from the TIL?

One of the major strengths of the study is the development of a unique peptide-exchangeable pHLA class I multimers. However, characterization and validation of these critical reagents were not sufficiently provided. At least the following points should be discussed: (i) efficiency of peptide-exchange reaction (% of monomers complexed with synthetic peptides) and a possibility that some of the tested epitopes may not be incorporated; (ii) stability of the reagents after loading with synthetic peptides and after multimerized with anti-His tag antibodies; and (iii) multiplexing ability, testing multiple epitopes using different fluorochromes in a single tube. Similarly, it will be important to discuss the possibility that the absence of T-cell response against the some tumor antigens is due to lack of expression of the antigens.

It is important to show tetramer-staining results for samples that are not considered to contain tumor antigen-specific T cells, for example, peripheral T cells from healthy donors. If low-frequency responses are detected in these negative control samples, further testing using aAPCs could be considered.

For experiments using overlapping peptides to identify novel epitopes in Figure 4, only successful results for MART-1, NY-ESO-1, and gp100 were described. If available, results for other antigens (SSX2, MAGE-A1, and survivin) are also important to report, even with the negative results.

[Editors' note: further revisions were suggested prior to acceptance, as described below.]

Thank you for resubmitting your work entitled "Landscape mapping of shared antigenic epitopes and their cognate TCRs of tumor-infiltrating T lymphocytes in melanoma" for further consideration by *eLife*. Your revised article has been evaluated by Jeffrey Settleman (Senior Editor) and a Reviewing Editor.

The manuscript has been improved but there are some remaining issues that need to be addressed before acceptance, as outlined below:

Reviewer No. 1 requests some textual changes and explanations that should be provided (points 1, 3, 4). The concern No. 1 by reviewer No. 2 should be commented on in the text, by discussing the limitation of the study with respect to the number of melanoma antigens explored in the work.

For your information, here are the reviewers' comments:

*Reviewer #1:*

The authors seem to have addressed many/most of the concerns from the previous reviews. I like the technology and feel like it will be useful, not sure how much the patent etc. is going to hinder its use elsewhere, but maybe the data can form a reference that others can check against.

Essential revisions:

1) In the abstract I am still uncomfortable with this statement "We were able to determine the specificity of 12.2 {plus minus} 7.3% (mean {plus minus} SD, max 25.9%, min 4.6%) of TILs recognizing 3.1 {plus minus} 2.0 (mean {plus minus} SD, max 7, min 1) previously known and novel epitopes across HLA-A, B, and C".

I think wording it like this

"On average we identified specificity of 12% of the TILs, and each TIL on average recognized 3 epitopes across various HLA-A, B and C alleles"makes the meaning clearer (and the uncertainty etc. can be given in the text) ?

2) The authors indicate they do not have PBMCs for the patients to check the "background" T cell population and how much is shared with the TILs. This seems like something that should not have happened? If so, much effort is being put into developing this technology and testing it, why handicap oneself at the start by going with samples that don't have the corresponding PBMCs? I am worried that this is an excuse to avoid addressing the question or not revealing "adverse" findings.

3) The term "exponential examination" of the immune response (line 76) might need to be explained. what aspect is exponential here?

4) The novelty of the technology: Introduction "We have developed a novel peptide-exchangeable pHLA class I multimer technology that can be applied for any given class I allele. Compared with conventional methods (Andersen et al., 2012), our multimers enable an exponential examination of the immune response, and are more effective in class I alleles that are difficult to produce in bacteria (Migueles et al., 2000)".

It was still not clear to me the exact novelty in the method. maybe a line or two could be added to explain the exact novelty and how it differs from other comparable technologies. Sorry for being difficult here, but I cannot still explain what is the novelty to myself after reading these lines (in addition to not knowing what the term "exponential examination" means).

*Reviewer #2:*

The authors have chosen an unusual approach to respond to the reviewers. Rather than going through each of the reviewer’s issues, they bundled them together and numbered them as comments 1-11.

Importantly, they chose to ignore all of my major points in this rebuttal. They only responded to one stylistic point (comment #1). All other points were ignored.

Essential revisions:

1) A significant limitation of this study is that the authors limited their effort to six previously defined shared melanoma antigens. Unfortunately, other types of antigens were not considered.

2) The authors also did not evaluate whether some of the T cells were neoantigen-specific, even though the tools were in hand. It would indeed be relevant to determine whether T cells specific for shared antigens or neoantigens are more common in TILs.

3) Overall, the scope of the paper is limited, given the primary focus of the study on the screening of TILs for their peptide specificity.

Overall, I believe that this paper is more suitable for a clinically oriented cancer immunology journal than *eLife*.

*Reviewer #3:*

The authors have satisfactorily addressed all my comments and concerns. This body of work is highly innovative and meritorious. It represents the most comprehensive testing of melanoma infiltrating T-cell reactivity against a large panel of shared tumor antigen-derived epitopes. The experimental approach and data interpretation are solid, and all appropriate controls were included or at least considered. The major conclusions are supported by the data.

---

## [Author Response]

Essential revisions:1) Figure 1 which shows the results from a screening experiment – in total this figure shows 200 FACS plots. The presentation of the data needs to be reconsidered to emphasize the key conclusions.

Thank you for the comment. According to the comment, we have modified how we present the data. We have improved the clarity and organization of the FACS plots in Figure 1. We now show the data of high-throughput pHLA multimer staining for B*07:02^+^ M68 TILs in Figure 1—figure supplement 1, while all the data of high-throughput staining for A*02:01^+^ and C*05:01^+^ M87 TILs and only the positive staining data for the other A*02:01^+^ TILs, A*24:02^+^ M37 TILs, and B*07:02^+^ M68 TILs are shown in Figure 1.

2) Sequencing the TCR repertoires to determine the actual clonality would have been very useful. But it does not seem to have been successful in most cases. Was it because the expanded clones were still polyclonal? A related question, how often are they polyclonal? This could be an interesting question to address.

Thank you for the important comment. We molecularly cloned TCR genes from multimer-positive T cells via 5’-rapid amplification of cDNA ends (RACE) PCR. All the HLA-B and C-restricted T cell populations contained one pair of TCR genes each. These results suggested that all the HLA-B and C-restricted antigen-specific T cells found in this study were monoclonal (Table 1).

We performed high-throughput TCR repertoire analysis for A*02:01-restricted CD8^+^ T cells specific to the A*02:01/MART1 (M25, M66, and M96), A*02:01/NY-ESO-1 (M31), A*02:01/SSX2 (M37 and M40), and A*02:01/gp100 (M66) multimers using CapTCR-seq (Mulder et al., Blood Adv. 2018). All their T cell populations contained polyclonal TCRα and TCRβ genes (data not shown).

Quite a few TCR genes recognizing A*02:01/MART1, NY-ESO-1, gp100, and tyrosinase have been cloned previously, and some of these TCRs have been tested in TCR gene therapy clinical trials (Feldman et al., 2015; Fesnak et al., 2016; Morgan et al., 2006). Therefore, we did not isolate TCR genes from multimer-positive T cells specific to these antigens. A*02:01/ABCB5-specific TCR genes were molecularly cloned from M87 TILs. Since ABCB5 is expressed in various normal tissues (https://www.proteinatlas.org/ENSG00000004846-ABCB5/tissue), TCR gene therapy targeting ABCB5 is very likely to cause undesired on-target off-tumor toxicity. Therefore, we did not study the tumor reactivity of the A*02:01/ABCB5 TCR.

Subsection “Primary T cells transduced with TIL-derived TCR genes recognize tumor cells”: “For the majority of tumor reactivities that we identified, multimer-positive antitumor T cells were collected, and their TCR genes were molecularly cloned. All the HLA-B- and HLA-C-restricted T cell populations contained one pair of TCR genes each. These results suggested that all the HLA-B- and HLA-C-restricted antigen-specific T cells found in this study were monoclonal (Table 1, Figure 5-supplement 1,2, Supplementary file 16).”

3) What about the obvious control, the circulating T cells from the patients, to see if they have a different characteristic from the TIL?

Thank you for the comment. We agree that the circulating T cells from the patients might have different characteristics from the TILs. Unfortunately, peripheral blood was not available from any of the patients, and we were not able to address the question.

4) One of the major strengths of the study is the development of a unique peptide-exchangeable pHLA class I multimers. However, characterization and validation of these critical reagents were not sufficiently provided. At least the following points should be discussed: (i) efficiency of peptide-exchange reaction (% of monomers complexed with synthetic peptides) and a possibility that some of the tested epitopes may not be incorporated; (ii) stability of the reagents after loading with synthetic peptides and after multimerized with anti-His tag antibodies; and (iii) multiplexing ability, testing multiple epitopes using different fluorochromes in a single tube. Similarly, it will be important to discuss the possibility that the absence of T-cell response against the some tumor antigens is due to lack of expression of the antigens.

Thank you for raising these important points. We have performed a number of experiments to evaluate the efficiency of the peptide-exchange reaction, stability, and multiplexing ability of the reagents.

(i) Efficiency of the peptide-exchange reaction in the monomers

We established a protocol to measure the efficiency of peptide exchange in soluble class I^Q115E^-K^b^ monomers and evaluated the efficiency for A*02:01, B*07:02, and C*07:02 with class I-restricted peptides (Supplementary files 2, 4, and 8). As shown in Figure 6—figure supplement 1, the peptide exchange efficiencies in the A*02:01, B*07:02, and C*07:02 monomers were 62.0 ± 14.9% (mean ± SD), 94.2 ± 12.0%, and 80.5 ± 22.1%, respectively. All the peptides tested in this experiment were efficiently exchanged in our monomers, except for one peptide (No. 9 in Figure 6—figure supplement 1C).

We have added this information to the Materials and methods section, as shown below.

Subsection “Measurement of peptide exchange efficiency using ELISA”:

“The efficiency of peptide exchange in soluble class I^Q115E^-K^b^ monomers was assessed using a competition binding assay and ELISA. The monomer was loaded with 100 μg/ml biotinylated peptide and incubated overnight at 37°C. Biotinylated peptide-monomer was purified and exchanged with PBS using Amicon Ultra filters (molecular weight cut-off (MWCO) 10 kDa) (Millipore Σ, Burlington, MA) and mixed with 1 mg/ml HLA-restricted peptide of interest or an equivalent volume of DMSO followed by overnight incubation at 37°C. ELISA plates were coated with anti-pan class I mAb (clone W6/32) at 10 μg/mL in PBS overnight at 4°C. The plates were washed and blocked with 10% nonfat dry milk in PBS for 30 minutes at room temperature. Peptide-monomer was added and incubated for 2 hours at room temperature. After washing, the plates were incubated with streptavidin-conjugated alkaline phosphatase for 30 minutes at room temperature. Finally, the plates were washed and incubated with p-nitrophenyl phosphate (PNPP) substrate (Pierce, Rockford, IL) at room temperature. The reaction was terminated by adding 1 mol/L NaOH. The optic density (OD) (405 nm) was read (Spectramax 190 Microplate Reader; Molecular Devices, Sunnyvale, CA). The OD values from the control wells containing nonbiotinylated peptide were subtracted from the OD values in test wells containing biotinylated peptide. The efficiency of peptide exchange for each monomer was calculated as follows: Peptide exchange efficiency = [1 – (OD value with peptide/OD value with DMSO)] x 100. Every sample was assayed in triplicate wells. The biotinylated peptides used were HLA-A*02:01-restricted telomerase_540-548_ ILAK(-biotin)FLHWL, B*07:02-restricted MiHAg SMCY_1041-1051_SPSVDK(-biotin)ARAEL, and C*07:02-restricted adenovirus B_585-593_FRK(-biotin)DVNMVL. The efficiencies of peptide exchange in the A*02:01, B*07:02, and C*07:02 monomers were shown in Figure 6—figure supplement 1.”

(ii) Stability of the reagents

We evaluated the stability of the reagents after peptide loading and multimerization. When the peptide-loaded monomers were multimerized with anti-His mAb at several time points, the ability of the A*02:01 and B*07:02 multimers to stain cognate T cells did not change for 5 days but decreased as time passed (Figure 6—figure supplement 2A). On the other hand, the ability of the pHLA multimers to stain T cells did not change until day 11 after multimerization (Figure 6—figure supplement 2B). These results suggested that the peptide-loaded class I monomer should be multimerized within 5 days and that the reagents after multimerization were stable for at least 10 days.

Subsection “Production of human cell-based pHLA multimers”: Soluble monomeric class I^Q115E^-K^b^ loaded with the peptide were multimerized using an anti-His mAb (clone AD1.1.10; Abcam) conjugated to a fluorochrome such as phycoerythrin (PE) or allophycocyanin (APC) at a 2:1 molar ratio for 2 hours at room temperature or overnight at 4°C.

(iii) Multiplexing ability of pHLA multimers

We first studied whether our pHLA complexes individually dimerized using anti-His mAb (clone AD1.1.10) conjugated with various fluorochromes (PE (Abcam), APC (Abcam), FITC (Abcam), PerCP (Abcam), Alexa Fluor 680 (Santa Cruz Biotechnology), Alexa Fluor 790 (Santa Cruz Biotechnology), and PE-Cy7 (Santa Cruz Biotechnology)), can stain cognate T cells. Although those dimerized with PE- or APC-conjugated anti-His mAb were similarly able to stain T cells, others were not (see Author response image 1). We then individually exchanged indicated HLA monomers with two different peptides, dimerized the peptide-exchanged monomers with PE- or APC-conjugated anti-His mAb, and used the dimers to stain TILs in a single tube. As shown in Figure 6—figure supplement 3, a pair of two pHLA multimers exchanged with different peptides and dimerized with different fluorochromes successfully stained two specific subsets of antigen-specific TILs in a single tube.

We have added the information to the Material and Methods section as shown below.

Subsection “Flow cytometry and cell sorting”: Cell surface molecules were stained with a PC5- or Pacific Blue-conjugated anti-CD8 mAb (clone B9.11; Beckman Coulter), FITC-conjugated anti-NGFR mAb (clone ME20.4; Biolegend), APC/Cy7-conjugated anti-CD3 mAb (clone UCHT1; Biolegend), and FITC-conjugated anti-HLA-A2 mAb (clone BB7.2; Biolegend).

Subsection “Production of human cell-based pHLA multimers”: Soluble monomeric class I^Q115E^-K^b^ loaded with the peptide was multimerized using an anti-His mAb (clone AD1.1.10; Abcam) conjugated to a fluorochrome such as phycoerythrin (PE) or allophycocyanin (APC) at a 2:1 molar ratio for 2 hours at room temperature or overnight at 4°C.

Subsection “pHLA multimer staining”: T cells (1 x 10^5^) were incubated for 30 minutes at 37°C in the presence of 50 nM dasatinib (LC laboratories). The cells were then washed and incubated with 5-10 μg/ml of PE-conjugated multimer for 30 minutes at room temperature, and R-phycoerythrin-conjugated AffiniPure Fab fragment goat anti-mouse IgG1 antibody (Jackson ImmunoResearch Laboratories) was added for 15 minutes at 4°C. Next, the cells were washed three times and costained with an anti-CD8 mAb for 15 minutes at 4°C. Dead cells were finally discriminated using the LIVE/DEAD Fixable Dead Cell Stain kit. For multiplex staining, after incubation with PE-conjugated multimer, the cells were washed three times and then incubated with 5-10 μg/ml of APC-conjugated multimer for 30 minutes at room temperature followed by costaining with an anti-CD8 mAb.

**Author response image 1. respfig1:** T cell staining using pHLA multimers with various fluorochrome-conjugated anti-His antibodies. A*02:01 monomer loaded with the heteroclitic NY-ESO-1_157-165_ peptide was multimelized with PE (Abcam), APC (Abcam), FITC (Abcam), PerCP (Abcam), Alexa Fluor 680 (Santa Cruz Biotechnology), Alexa Fluor 790 (Santa Cruz Biotechnology), or PE-Cy7 (Santa Cruz Biotechnology)-conjugated anti-His mAb (clone AD1.1.10), respectively, and Jurkat 76/CD8 cells transduced with A*02:01/NY-ESO-1_157-165_ TCR (clone 1G4LY) were stained with their pHLA multimers. The A*02:01/MART1_27-35_ TCR (clone DMF5) was used as a control. The percentages of multimer^+^ CD8^+^ T cells are shown.

Subsection “pHLA multimer staining”: The stability and multiplexing ability of the reagents were confirmed (Figure 6—figure supplement 2,3).

(iv) Similarly, it will be important to discuss the possibility that the absence of T-cell response against some tumor antigens is due to lack of expression of the antigens.

Thank you for this important comment. We added the sentences below to the Results.

Subsection “T cell epitope determination of in vitro-expanded TILs using overlapping peptides”: It should be noted that in this study, we did not investigate the expression of shared antigens in the patient’s own tumor tissues. Therefore, it is possible that the absence of a T cell response toward some shared antigens is due to the lack of their expression.

5) It is important to show tetramer-staining results for samples that are not considered to contain tumor antigen-specific T cells, for example, peripheral T cells from healthy donors. If low-frequency responses are detected in these negative control samples, further testing using aAPCs could be considered.

Thank you for the comment. In the revised manuscript, we stained peripheral T cells from healthy donors with pHLA multimers that positively stained TILs. CD3^+^ T cells were purified from HLA class I-matched healthy donors and stained with the indicated pHLA multimers. HLA-B*40:02^+^/41:01^+^ healthy donor T cells were used for B*40:01 multimer staining since B*40:01^+^ healthy donor T cells were not available.

It is well known that MART1, gp100, and tyrosinase are tumor-differentiation antigens expressed by normal melanocytes and that healthy normal donors as well as melanoma patients have A*02:01-restricted CD8^+^ T cells specific to these antigens in the periphery. Therefore, we did not perform multimer staining for these antigens.

As in Author response image 2/2a, antigen-specific T cells specific to the epitope peptides identified in this study were not found in peripheral T cells from HLA-matched healthy donors.

**Author response image 2. respfig2:** Staining of peripheral T cells from healthy donors with pHLA multimers. Peripheral T cells from HLA-matched healthy donors were stained with pHLA multimers that positively stained TILs for HLA-A*02:01^+^ (**A**), A*24:02^+^ (**B**), B*07:02^+^ (**C**), B*18:01^+^ (**D**), B*40:02^+^ (**E**), B*41:01^+^ (**E**), C*03:04^+^ (**F**), C*05:01^+^ (**G**), C*06:02^+^ (**H**), and C*07:01^+^ (**I**) healthy donors. The A*02:01/Flu MP_58-66_ (**A**), A*24:02/EBV LMP2_222-230_ (**B**), and B*07:02/EBV EBNA3A_379-387_ (**C**) multimers were employed as positive controls. The A*02:01/HTLV-1 tax_11-19_ (**A**), A*24:02/HTLV-1 tax_301-309_ (**B**), B*07:02/HIV nef_128-137_ (**C**), B*18:01/HIV gag_161-170_ (**D**), B*40:01/HIV nef_92-100_ (**E**), C*03:04/HIV gag_164-172_ (**F**), C*05:01/HIV rev_67-75_ (**G**), C*06:02/HIV nef_120-128_ (**H**), and C*07:01/HIV nef_105-115_ (**I**) multimers were used as negative controls. The percentage of multimer^+^ cells in CD8^+^ T cells is shown.

For experiments using overlapping peptides to identify novel epitopes in Figure 4, only successful results for MART-1, NY-ESO-1, and gp100 were described. If available, results for other antigens (SSX2, MAGE-A1, and survivin) are also important to report, even with the negative results.

Thank you for the comment. Although we performed similar experiments using overlapping peptides for SSX2, MAGE-A1, and survivin, no T cell response was observed in the TILs.

Subsection “T cell epitope determination of in vitro-expanded TILs using overlapping peptides”: Although we performed similar experiments using overlapping peptides for SSX2, MAGE-A1, and survivin, no T cell response was observed.

[Editors' note: further revisions were suggested prior to acceptance, as described below.]

The manuscript has been improved but there are some remaining issues that need to be addressed before acceptance, as outlined below:Reviewer #1:The authors seem to have addressed many/most of the concerns from the previous reviews. I like the technology and feel like it will be useful, not sure how much the patent etc. is going to hinder its use elsewhere, but maybe the data can form a reference that others can check against.Essential revisions:1) In the abstract I am still uncomfortable with this statement "We were able to determine the specificity of 12.2 {plus minus} 7.3% (mean {plus minus} SD, max 25.9%, min 4.6%) of TILs recognizing 3.1 {plus minus} 2.0 (mean {plus minus} SD, max 7, min 1) previously known and novel epitopes across HLA-A, B, and C".I think wording it like this"On average we identified specificity of 12% of the TILs, and each TIL on average recognized 3 epitopes across various HLA-A, B and C alleles"makes the meaning clearer (and the uncertainty etc. can be given in the text?

We apologize for making reviewer 1 uncomfortable with the statement. We propose to modify the sentence as below.

"We were able to determine the specificity, on average, of 12.2% of the TILs recognizing a mean of 3.1 shared antigen-derived epitopes across HLA-A, B, and C."

3) The term "exponential examination" of the immune response (line 76) might need to be explained. what aspect is exponential here?

We agree that the word, “exponential” was confusing. We have rewritten the sentences as shown below in our response to Comment 4).

4) The novelty of the technology: Introduction "We have developed a novel peptide-exchangeable pHLA class I multimer technology that can be applied for any given class I allele. Compared with conventional methods (Andersen et al., 2012), our multimers enable an exponential examination of the immune response, and are more effective in class I alleles that are difficult to produce in bacteria (Migueles et al., 2000)".It was still not clear to me the exact novelty in the method. maybe a line or two could be added to explain the exact novelty and how it differs from other comparable technologies. Sorry for being difficult here, but I cannot still explain what is the novelty to myself after reading these lines (in addition to not knowing what the term "exponential examination" means).

We apologize that these sentences were not clear enough to explain the novelty of our technologies. We have rewritten the sentences as shown below.

“We have developed a novel peptide-exchangeable pHLA class I multimer technology that can be applied for any given class I allele and bound peptide. Unlike comparable methods (Migueles et al., 2000; Andersen et al., 2012), our novel pHLA multimer technology enables a one-step peptide exchange in one tube and is more effective in class I alleles that are difficult to produce in bacteria. These technical advantages allow for a rapid, versatile, and less labor-intensive examination of the immune response. Our strategy using a paired library of the novel pHLA multimer and human cell-based artificial APC system enables identification of novel class I epitopes and detection of cognate T cells in a comprehensive and systematic way.”Reviewer #2:Essential revisions:1) A significant limitation of this study is that the authors limited their effort to six previously defined shared melanoma antigens. Unfortunately, other types of antigens were not considered.

We apologize for the confusion. In addition to 304 peptides derived from the six previously defined shared melanoma antigens (Supplementary file 2), we studied 501 peptides derived from 91 additional proteins (Supplementary file 1). Accordingly, we studied greater than 800 peptides derived from more than 90 proteins overall. As reviewer 3 pointed out, “It represents the most comprehensive testing of melanoma infiltrating T-cell reactivity against a large panel of shared tumor antigen-derived epitopes.” And yet, we absolutely agree with the reviewer 2 in that our study is still limited and additional comprehensive studies are required to elucidate the full spectrum of antitumor T cell response in TILs. We have added the sentences to Discussion section.

“Although we studied T cell responses to greater than 800 peptides derived from more than 90 proteins across all the 25 class I alleles expressed by 8 melanoma patients, our study is still limited. Additional comprehensive studies are required to elucidate the full spectrum of antitumor T cell response in TILs.”